# Soil moisture estimation in South Asia via assimilation of SMAP retrievals

Jawairia A. Ahmad[1], Barton A. Forman[1], and Sujay V. Kumar[2]

[1]Department of Civil and Environmental Engineering, University of Maryland, College Park, MD, USA
[2]Hydrological Sciences Laboratory, NASA Goddard Space Flight Center, Greenbelt, MD, USA

**Correspondence:** Jawairia A. Ahmad (jahmad@umd.edu)

**Abstract.** A soil moisture retrieval assimilation framework is implemented across South Asia in an attempt to improve regional soil moisture estimation as well as to provide a consistent regional soil moisture dataset. This study aims to improve the spatiotemporal variability of soil moisture estimates by assimilating Soil Moisture Active Passive (SMAP) near-surface soil moisture retrievals into a land surface model. The Noah-MP (v4.0.1) land surface model is run within the NASA Land Information System software framework to model regional land surface processes. NASA Modern-Era Retrospective Analysis for Research and Applications (MERRA2) and GPM Integrated Multi-satellitE Retrievals (IMERG) provide the meteorological boundary conditions to the land surface model. Assimilation is carried out using both cumulative distribution function (CDF) corrected (DA-CDF) and uncorrected SMAP retrievals (DA-NoCDF). CDF-matching is applied to correct the statistical moments of the SMAP soil moisture retrieval relative to the land surface model. Comparison of assimilated and model-only soil moisture estimates with publicly-available in-situ measurements highlight the relative improvement in soil moisture estimates by assimilating SMAP retrievals. Across the Tibetan Plateau, DA-NoCDF reduced the mean bias and RMSE by 8.4% and 9.4%, even though assimilation only occurred during less than 10% of the study period due to frozen (or partially frozen) soil conditions. The best goodness-of-fit statistics were achieved for the IMERG DA-NoCDF soil moisture experiment. The general lack of publicly available in-situ measurements across irrigated areas limited a domain-wide direct model validation. However, comparison with regional irrigation patterns suggested correction of biases associated with an unmodeled hydrologic phenomenon (i.e., anthropogenic influence via irrigation) as a result of SMAP soil moisture retrieval assimilation. The greatest sensitivity to assimilation was observed in cropland areas. Improvements in soil moisture potentially translate into improved spatiotemporal patterns of modeled evapotranspiration, although limited influence from soil moisture assimilation was observed on modeled processes within the carbon cycle such as gross primary production. Improvement in fine-scale modeled estimates by assimilating coarse-scale retrievals highlights the potential of this approach for soil moisture estimation over data scarce regions.

# 1 Introduction

Soil moisture (SM) is an important variable in geophysical science. In land surface models, soil moisture primarily influences the energy cycle by controlling latent heat flux and soil temperature (Al-Kayssi et al., 1990), and the water cycle via evapotranspiration, soil infiltration capacity, and runoff (Penna et al., 2011). Accurate soil moisture estimation is also a requirement for analyzing the effects of climate change as soil moisture variability influences terrestrial carbon uptake (Green et al., 2019). In the context of agriculture, soil moisture provides a quantitative basis for the development of sociopolitical policies aimed at regulating and monitoring crop cultivation, crop selection, water resources distribution, and irrigation processes (Schneider, 1989; Shani et al., 2004; Jalilvand et al., 2019). Soil moisture-based frameworks have been extensively used for irrigation scheduling and monitoring, particularly in terms of tracking plant growth (Dukes and Scholberg, 2005; Soulis et al., 2015). The three main sources of surface soil moisture are precipitation (Morin and Benyamini, 1977; Douville et al., 2001; Wei and Dirmeyer, 2012), freshwater flow across the floodplain (Daly and Porporato, 2005), and surface irrigation via groundwater pumping. The feedback loop between soil moisture and each of these sources varies in space and time according to the geographic and topological features of the locale (Wei and Dirmeyer, 2012).

Various techniques have been used for soil moisture characterization such as in-situ station networks, physical modeling, and remote sensing (Seneviratne et al., 2010; Hauser et al., 2017; Reichle et al., 2021). While the in-situ station data is considered most representative of the true ground conditions, it is generally limited by data sparsity and data availability. In contrast, physical modeling can be leveraged to provide estimates at fine spatiotemporal resolutions. However, contemporary modeling techniques lack comprehensive representation of the complex relationships between all geophysical variables. Remote sensing has also been widely utilized in soil moisture estimation to translate optical (Piles et al., 2011), thermal infrared (Zhang et al., 2014), and microwave (Entekhabi et al., 2010; Panciera et al., 2013) observations into soil moisture retrievals.

While providing useful information, remote sensing-based soil moisture retrievals are limited by the accuracy of the retrieval algorithm, swath width, field-of-view, and the orbital specifications of the observing instrument onboard the satellite. One effective method for overcoming some of the limitations posed by physical modeling and remote sensing sensors is data assimilation. Data assimilation (DA) is a technique used to merge modeled estimates with observations while taking into consideration their respective errors and uncertainties (Kalman, 1960; McLaughlin, 2002). The posterior estimate obtained through DA combines the strengths of both models and observations to yield a dataset that is improved relative to the standalone products (Zhang and Moore, 2015). Several studies have attempted to improve water budget estimation by assimilating soil moisture information into a land surface model (LSM). Huang et al. (2008) assimilated in-situ surface soil moisture measurements and low-frequency passive microwave remote sensing data into the Simple Biosphere Model (SiB2) and produced improvements in

surface soil moisture estimates. Lievens et al. (2015) modeled the hydrologic cycle over the Murray Darling Basin in Australia and explored the results of assimilating Soil Moisture and Ocean Salinity (SMOS) soil moisture retrievals into the Variable Infiltration Capacity (VIC) model. They concluded that improvements in the wetness conditions due to soil moisture retrieval assimilation translated into improved predictions of associated water fluxes. Comparison of modeled soil moisture estimates with soil moisture retrievals revealed an inherent bias in the statistical moments of the studied retrievals (Reichle et al., 2004). A bias correction technique based on CDF-matching suggested by Reichle and Koster (2004) demonstrated better conformity in the statistical moments between the LSM soil moisture estimates and the satellite-based soil moisture retrievals. However, Kumar et al. (2015) showed that retrieval distribution mapping via CDF-matching could result in the removal of information pertaining to the irrigation signal. Nearing et al. (2018) also attributed loss of signal information to CDF-matching during data assimilation.

According to the current climate change forecasts, severe water stress is predicted in various parts of South Asia (Sivakumar and Stefanski, 2010). Total groundwater storage in northwestern India has undergone a decline, which is likely linked to irrigation-induced groundwater pumping (Rodell et al., 2009; Asoka et al., 2017). Global land surface models, in general, do not include groundwater pumping modules. An inverse technique of estimating the amount of groundwater pumped could potentially be developed if accurate soil moisture estimates are available (apart from the other water budget contributing variables). Soil moisture records may be able to provide much needed information about the extent and amount of groundwater pumping across the whole of South Asia. Accurate soil moisture estimation across South Asia is, therefore, an important need.

In-situ soil moisture measurements across South Asia are sparse at best. To fill this knowledge gap and to evaluate the utility of leveraging data assimilation as a feasible option in this region, we demonstrate the utilization of Soil Moisture Active Passive (SMAP; Entekhabi et al. (2010)) retrieval assimilation to improve soil moisture estimation across South Asia and the adjoining mountainous areas. Section 2 describes the prominent features of the study domain; Sect. 3 provides details regarding the various datasets and the data assimilation framework utilized; Sect. 4 highlights the important results of the DA experiments; Sect. 5 includes a discussion of the salient findings of the DA experiments, and Sect. 6 summarizes the main conclusions of this study.

## 2 Study domain

The study domain discussed in this paper encompasses the mountainous region in South Asia and the adjoining areas, Fig. 1. The HinduKush-Himalayan mountain range and the Tibetan Plateau, represented by grid cells with elevation > 2000 m in Fig. 1(a) constitute high mountain Asia. Ten major rivers (Indus, Ganges, Brahmaputra, Irrawaddy, Salween, Mekong, Yangtze,

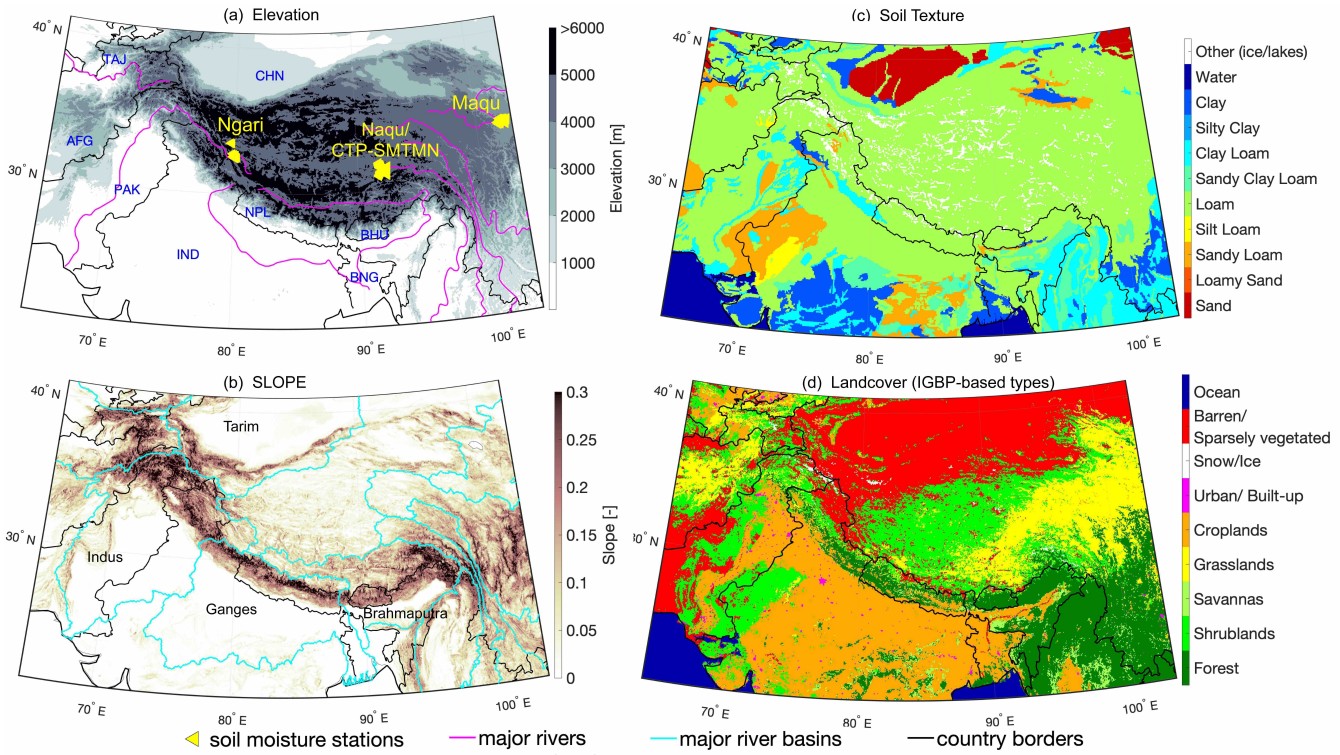

**Figure 1.** The study domain encompasses the mountainous region in South Asia and the adjoining areas. High elevation and high slope (>0.2) areas demarcate the HinduKush-Himalayan mountain range, whereas the high elevation and mild slope (<0.1) grid cells demarcate the Tibetan Plateau (subplots (a) and (b)). The yellow markers in subplot (a) locate the stations within the Tibetan Plateau used to evaluate the soil moisture estimates (Sect. 4.1). The domain soil texture was categorized into 11 soil types (subplot c) according to the NCEP/STATSGO+FAO classification. The domain landcover comprised 10 main types based on the MODIS-derived IGBP classification (subplot d). AFG= Afghanistan, BNG = Bangladesh, BHU= Bhutan, CHN= China, IND= India, NPL = Nepal, PAK= Pakistan, TAJ= Tajikistan.

Yellow, Tarim, Amu, and Syr) originate in this region and flow towards the low elevation areas where they serve as sources of freshwater for the residing populace. Agriculture-based irrigation is a primary consumer of the freshwater transported downstream by the rivers (Wester et al., 2018).

Figure 1(c) and Table A1 present the soil texture conditions within the domain. The NCEP/STATSGO+FAO (Natural Resources Conservation Service) soil texture classification is used to categorize the grid cells into 16 individual classes (Note: soil classes that did not have any grid cell types in the study domain are excluded from the figure legend). The predominant soil texture type found within the domain is loam followed by clay loam. Landcover categorization (see Fig. 1(d) and Table A1 columns 4 to 6) is based on the NCEP/MODIS-based International Geosphere–Biosphere Programme (IGBP) (Friedl et al., 2002) classification (Note: similar classes are lumped together, for example different forest types are grouped into a singular forest class). The predominant landcover types present within the study domain are barren, croplands, and shrublands.

The Food and Agriculture Organization (FAO) of the United Nations provides a global map of fraction areas that are

equipped for irrigation as part of the Global Map of Irrigation Areas (GMIA) product, which is provided at a 5-arc minute (0.0833°) resolution (Siebert et al., 2005). The GMIA product was used in this study to represent the total irrigation-equipped area within each grid cell, see Fig. 6(c). The grid cells with high irrigated percentages correspond well (spatially) with grid cells belonging to the landcover type croplands in Fig. 1(d).

## 3    Methodology and datasets

This section describes the methodology developed to implement the assimilation of SMAP soil moisture retrievals into the land surface model as well as the various datasets utilized in the analysis results detailed in Sect. 4.

### 3.1    NASA Land Information System

The NASA Land Information System (LIS) is a software framework that facilitates high performance computing for land surface modeling and data assimilation purposes (Kumar et al., 2006; Peters-Lidard et al., 2007). The NASA LIS framework

was used to run the Noah-MP land surface model and to assimilate SMAP soil moisture retrievals (Fig. 2).

#### 3.1.1    Noah-MP land surface model

Noah-MP (version 4.0.1) (Ek et al., 2003; Niu et al., 2011; Yang et al., 2011) was run within LIS to simulate the relevant land surface processes across the study domain. Noah-MP was run on an equidistant cylindrical grid with a spatial resolution of 0.05° x 0.05° at a 15 minute timestep. Table 1 outlines the Noah-MP configurations applied in this study.

Noah-MP was selected for this study due to the multilayer representation of soil, explicit modeling of frozen soil permeability (Niu and Yang, 2006), and representation of snowpack and soil interface processes. Noah-MP includes coupled energy, water, and carbon cycle simulation routines. The soil profile is divided into four layers with thicknesses of 5, 35, 60, and 100 cm, respectively. Updates in the surface soil moisture information are propagated to the underlying soil layers based on the water diffusivity and hydraulic conductivity, maximum moisture threshold of soil layers, and moisture flux between subsequent layers

of the soil. Noah-MP connects subsequent soil layers such that excessive water above saturation in a layer is moved to the next unsaturated layer. A three-layer (maximum) snow structure is implemented above the surface soil layer to capture snowpack dynamics and the snowpack-soil interface fluxes for areas that experience snowfall (Niu et al., 2011). Noah-MP was forced with meteorological fields from Modern-Era Retrospective analysis for Research and Applications (MERRA2, Gelaro et al. (2017)) and Integrated Multi-satellite Retrievals for Global Precipitation Measurement (IMERG, Huffman et al. (2015)). The

**Table 1.** Selection of model components in Noah-MP version 4.0.1 as implemented within LIS (Sect. 3.1.1).

| Model Components | Selected Inputs or Parameterizations |
| --- | --- |
| Elevation, slope, and aspect | SRTM30-v2.0 (Farr et al., 2007) |
| Landcover | MODIS (IGBPNCEP) (Friedl et al., 2002) |
| Maximum albedo | National Centers for Environmental Prediction (Robinson and Kukla, 1985) |
| Greenness | National Centers for Environmental Prediction (Gutman and Ignatov, 1998) |
| Vegetation | Dynamic vegetation option |
| Canopy stomatal resistance | Ball-Berry method (Ball et al., 1987) |
| Runoff and groundwater | Simple groundwater model, SIMGM (Niu et al., 2007) |
| Supercooled liquid water and frozen soil permeability | NY06 (Niu and Yang, 2006) |
| Surface-layer drag coefficient | General Monin-Obukhov similarity theory (Brutsaert, 2013) |
| Snow surface albedo | Biosphere-Atmosphere Transfer Scheme (Yang and Dickinson, 1996) |
| Partitioning of rain and snowfall | Jordan91 (Jordan, 1991) |
| Snow and soil temperature | Semi-implicit option |
| Lower boundary of soil temperature | Noah native option |
| Meteorological boundary conditions | MERRA-2 (Gelaro et al., 2017), IMERG (Huffman et al., 2015) |

IMERG Final run was used. It is important to note here that external irrigation and groundwater pumping were not explicitly modeled in Noah-MP. Thus, there was an information gap regarding these two water sources in the modeled water cycle.

## 3.2  Data sets

### 3.2.1  SMAP Level3 soil moisture for assimilation

Soil Moisture Active Passive (SMAP) is a satellite mission that follows a near-polar, sun-synchronous, 8-day repeat orbit (O'Neill et al., 2014). The L3SMP Level-3 soil moisture product is utilized in this study. It consists of daily estimates of global soil moisture within the top ∼5 cm as retrieved by the SMAP passive microwave L-band radiometer (O'Neill et al., 2019). The soil moisture data are provided on a global, cylindrical 36 km Equal-Area Scalable Earth Grid, Version 2.0 (Brodzik et al., 2012) beginning from 31 March 2015.

L-band radiometry offers all-weather, diurnal sensing of the surface dielectric properties. The surface dielectric properties are a function of the near-surface soil moisture. Several mitigation features directed at preventing signal contamination due to radio frequency interference (RFI) are built into the radiometer electronics and algorithms. Quality flags are included in the metadata to provide location specific details such as retrieval error, retrieval uncertainty, frozen ground conditions, presence of steep topography, and forest coverage (O'Neill et al., 2019).

### 3.2.2 In-situ soil moisture measurements for model evaluation

Ground-based soil moisture measurements were obtained from the International Soil Moisture Network, an international, multi-agency cooperation that provides global, in-situ soil moisture measurements for the validation of model and remote sensing-based products (URL= https://ismn.earth/en/). Station measurements from four separate networks: 1) Ngari, 2) Naqu, 3) Maqu (Su et al., 2011; Zeng et al., 2016), and 4) CTP-SMTMN (Yang et al., 2013) were colocated with the land surface model grid for evaluation of the modeled estimates. The colocation was based on a simple arithmetic averaging of stations 135 located within each grid cell.

The different networks represent varying local climates, although all networks are located at high elevations and have relatively cold climates. The Ngari network is located in an arid region, Naqu and CTP-SMTMN networks are situated in a semiarid region, and Maqu experiences a relatively humid climate, Fig. 1(a). The total number of stations available for evaluation is 101. Soil moisture measured at a depth of 5 cm below the surface was compared with model estimated surface soil moisture (soil 140 layer depth = 0 to 5 cm). Measurements across the Tibetan Plateau are the only publicly-available soil moisture measurements within the study domain between the years 2015 to 2020.

### 3.2.3 ALEXI evapotranspiration for model evaluation

To study the influence of soil moisture assimilation on related geophysical fluxes, the Atmosphere-Land Exchange Inverse (ALEXI) evapotranspiration product was used. ALEXI estimates evapotranspiration (ET) using multi-sensor thermal infrared 145 observations (Anderson et al., 2007, 2011). A two-source (soil and canpoy) land surface model is coupled to an atmospheric boundary layer model in order to derive energy fluxes based on thermal imagery and insolation estimates derived from geostationary satellites. The thermal infrared information-driven surface energy balance model takes vegetation cover (obtained from Moderate-resolution Imaging Spectroradiometer (MODIS) based normalized difference vegetation index) and the change in radiometric temperature of the land surface as inputs and estimates sensible, latent and ground heat fluxes as well as evapo-150 transpiration. ET estimates are provided at 0.05° x 0.05° spatial resolution at a daily temporal scale.

### 3.2.4 FluxSat gross primary production for model evaluation

FluxSat is a satellite-based product that employs machine learning, reflectance data from MODIS, and eddy covariance measurements to estimate global gross primary production (Joiner and Yoshida, 2020). Gross primary production, or GPP, is an important variable within the carbon cycle. It represents the rate at which carbon is assimilated into the plant biomass per unit 155 area per time during photosynthesis (Gough, 2011). GPP impacts the water cycle as plants transpire water during photosyn-

thesis, thereby acting as moisture sources for the atmosphere and moisture sinks within the soil (Philander, 2008). FluxSat is developed by training neural networks using MODIS reflectance data to upscale GPP obtained from eddy covariance flux tower measurements (Joiner and Yoshida, 2020). FluxSat GPP was used here to study the influence of soil moisture assimilation on the carbon cycle.

### 3.2.5 GOME-2 fluorescence for model evaluation

In addition to GPP from FluxSat, solar-induced fluorescence (SIF) retrievals were also utilized to investigate the influence of soil moisture assimilation on the resulting carbon flux. Joiner et al. (2013) retrieved chlorophyll fluorescence using observations near the 740 nm emission peak gathered by the Global Ozone Monitoring Experiment 2 (GOME-2) spectrometer aboard the European meteorological (MetOp) satellites. Satellite-based fluorescence retrievals can be exploited to infer the functional status of vegetation (Van der Tol et al., 2014). Chlorophyll-excitation induced by solar energy results in fluorescence generated during photosynthesis. Carbon is then taken in by vegetation during photosynthesis. Considering the link to photosynthesis, Joiner et al. (2014) used SIF as an analog for GPP and highlighted the conformity within their phenologic responses. Joiner et al. (2014) also examined the seasonal cycles of modeled GPP in conjunction with GOME-2 fluorescence retrievals to track seasonal patterns in photosynthesis. The GOME-2 satellite fluorescence data is available at a spatial resolution of $0.5°$ x $0.5°$ at a monthly time scale and includes estimated errors on the order of 0.1–0.4 mW $m^{-2}$ $nm^{-1}$ $sr^{-1}$.

### 3.3 Experimental framework

Three types of model runs were implemented in LIS, Fig. 2. Details of each of the three types of model runs are provided below.

### 3.3.1 Nominal replicate (NR)

Noah-MP was run for five years from 1 January 2010 to 31 December 2014 using a single, nominal replicate (NR) to provide initial soil moisture conditions for the Open Loop and data assimilation runs (discussed further below). The NR simulation was also utilized to develop the model cumulative distribution functions (CDFs) that were later used for CDF-matching during the assimilation run discussed in Sect. 3.3.3.

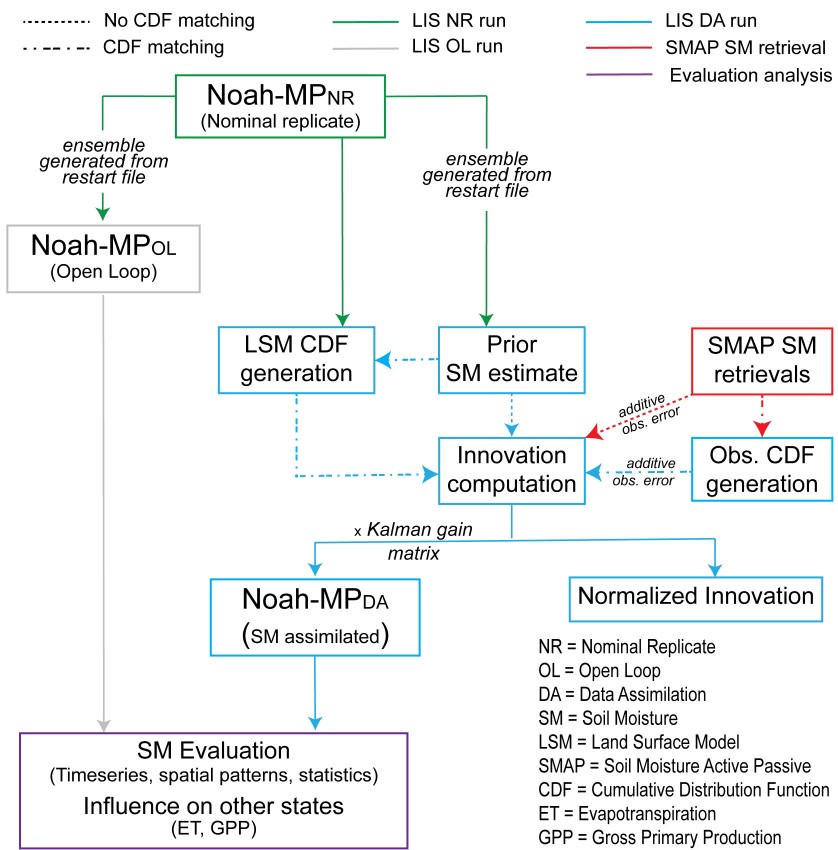

**Figure 2.** Overview of the soil moisture assimilation methodology implemented within the NASA Land Information System using the Ensemble Kalman Filter.

### 3.3.2 Open loop (OL)

The OL run represents a model-only run, i.e., the Noah-MP model was run in an ensemble configuration without any external observations assimilated. The OL run serves as a baseline for Noah-MP's land surface modeling capability across South Asia for eventual comparison with the DA run detailed in Sect. 3.3.3.

    The NR restart file provided the initial conditions for the OL run which started on 1 January 2015 and extended to 30 September 2020 (Fig. 2). The NASA Land Data Toolkit (LDT; Arsenault et al. (2018)) was used to upscale to a 20-replicate

ensemble from the single replicate NR restart file. The number of replicates was selected through an ensemble analysis; the ensemble standard deviation (as a function of time) was studied as the number of replicates in the ensemble was increased from five to 50. It was found that as the number of replicates increased beyond 15, the ensemble standard deviation reached

an asymptotic value. Therefore, a 20-replicate ensemble was selected to represent a low-rank approximation of the probability distribution that reasonably captures the true uncertainty in the model estimates.

Boundary conditions such as air temperature and radiative fluxes (i.e., incident shortwave and longwave radiation) were provided by MERRA2. Boundary condition (forcing) perturbations used by Kwon et al. (2019) were applied while propagating the ensemble forward in time, see Table B1. Two different sets of precipitation datasets were used to drive Noah-MP: i) MERRA2 (Gelaro et al., 2017), and ii) GPM IMERG (Huffman et al., 2015). Usage of two different boundary condition (precipitation) sources was motivated by efforts to differentiate between the influence of model physics versus boundary conditions on the

prognostic variables, e.g., soil moisture. Comparison of the results obtained from MERRA2-forced versus IMERG-forced OL and DA experiments aided in understanding the influence of boundary conditions and the effect of SMAP retrieval assimilation on model SM estimation.

    The OL simulation from 1 January 2015 to 30 September 2015 served as the model ensemble spin-up to achieve realistic uncertainty in soil moisture estimates. The results detailed in Sect. 4 are computed from the OL and DA experiments for water

200 years 2016 to 2020. The water year demarcation used in this study starts in October of the preceding year (e.g., 2015) and ends in September of the relevant year (e.g., 2016).

### 3.3.3   Data assimilation (DA)

SMAP SM retrievals are available from 31 March 2015 onwards. In accordance with the availability of SMAP retrievals, the DA run started on 1 April 2015 and extended to 30 September 2020. The ensemble Kalman filter (EnKF) assimilation algorithm

was utilized to assimilate the SMAP SM retrievals into the Noah-MP modeled estimates.

    The EnKF algorithm consists of two main steps: i) propagation step, and ii) update step. Noah-MP is the non-linear forward model used to propagate the prognostic state vector ($y_t$) forward in time as $y_t(x) = f(y_{t-1}(x), \alpha)$, where $f(\cdot)$ is the Noah-MP model, $\alpha$ is a vector of model parameters, $t$ is time, and $x \in$ X defines the spatial domain. Equation (1) defines the formulation of the update step applied to the *a priori* state estimate (for each replicate) based on the difference between the model estimate

and the observed value:

$$y_t^+(x) = y_t^-(x) + K_t(x) \left( z_t(x) + v_t - H(y_t^-(x)) \right) \tag{1}$$

where $K_t(x) = C_{y_t z_t}(x) [C_{z_t z_t}(x) + C_{vv}]^{-1}$                    (2)

such that $y_t^+(x)$ = *a posteriori* soil moisture value at time $t$, $y_t^-$ = *a priori* soil moisture estimate at time $t$, $K_t(x)$ = Kalman
gain at time $t$, $z_t(x)$ = SMAP soil moisture retrieval at time $t$, $v_t$ = SMAP soil moisture retrieval error at time $t$ such that
$v_t \sim \mathcal{N}(0, \sigma_{vv}^2)$, $H(.)$ is the linear observation operator, $C_{y_t z_t}(x)$ = time-varying cross-covariance matrix between the *a priori*
state errors and the predicted observation errors, $C_{z_t z_t}(x)$ = time-varying predicted observation error covariance, and $C_{vv}$ =
time-invariant SMAP soil moisture retrieval error covariance.

The difference between the observation (plus observation error) and the mapped *a priori* model state estimate is known as the
innovation, $In_t$. The normalized innovation ($NI_t$) is an effective diagnostic tool that aids in the diagnosis of the assimilation
framework and the origin of biases (Buehner, 2010). Equation (3) provides the normalized innovation formula for each replicate
as:

$$
NI_t(x) = \frac{z_t(x) + v_t - H(y_t^-(x))}{\sqrt{C_{z_t z_t}(x) + C_{vv}}} \tag{3}
$$

The numerator in Eq. (3) equals $In_t$ which is then normalized by the squared-root of the sum of $C_{z_t z_t}$ and $C_{vv}$. In an optimal
DA system, the normalized innovations should exhibit a standard normal distribution ($NI_t \sim \mathcal{N}(0,1)$). To compute $C_{z_t z_t}$ and
$C_{vv}$, the prognostic state and observation error standard deviation was taken equal to 0.04 m$^3$ m$^{-3}$ (O'Neill et al., 2014). Test
simulations were conducted to ascertain the most suitable model and SMAP soil moisture retrieval error values (results not
shown). Model error standard deviation was increased from 0.02 m$^3$ m$^{-3}$ to 0.10 m$^3$ m$^{-3}$, while the SMAP error standard
deviation was kept fixed at the standard value used in literature, i.e., 0.04 m$^3$ m$^{-3}$. Similarly, the model error was fixed while
the SMAP soil moisture error standard deviation was increased. Based on the test results, it was noted that the smallest bias
and RMSE values were achieved for model and SMAP soil moisture retrieval error standard deviations equal to 0.04 m$^3$ m$^{-3}$.
It is worth noting here that the EnKF is expected to behave suboptimally given the nonlinearity of the Noah-MP model in
conjunction with the non-Gaussianity of the SMAP retrieval errors. However, the exploration of $NI_t$ sequence is a worthwhile
exercise in an effort to better diagnose the behavior of the assimilation framework used in this study.

As part of the experimental matrix, the DA experiments were implemented using two different approaches. First, a cumu-
lative distribution function (CDF) matching technique (Reichle and Koster, 2004) was used for bias correction of the SMAP
soil moisture retrievals, herein referred to as DA-CDF. Monthly CDFs of the SMAP soil moisture retrievals and the Noah-MP
modeled SM were developed using the NASA Land Data Toolkit. The monthly CDFs were then used to map the SMAP SM
retrievals into the Noah-MP modeled soil moisture space prior to assimilation. The second approach employed no bias correc-
tion applied to the SMAP SM retrievals using CDF-matching and the *raw* SMAP SM was assimilated into Noah-MP, herein
referred to as DA-NoCDF. The relative systematic errors between SMAP SM and modeled Noah-MP SM are ignored during

DA-NoCDF runs. Since the SMAP retrievals being assimilated represent the top ~5 cm of surface soil, the soil moisture in the topmost soil layer is the model state variable considered during assimilation. The OL and DA runs were then compared against the evaluation datasets to analyze the influence of SM assimilation on the modeled states in Sect. 4.

## 4   Experimental results

Model estimates for water years (October to September) 2016 to 2020 are used to compute the results presented in this section. Water years were used rather than Julian years due to the former's hydrologic suitability for the state variable under consideration, i.e., soil moisture (SM).

### 4.1   Evaluation using in-situ measurements

In-situ SM measurements available across the Tibetan Plateau were used to evaluate the modeled SM estimates. In-situ measurements were collected at the point-scale whereas the Noah-MP grid size equaled 0.05° x 0.05° (~5.5 km x ~5 km at mid-latitudes). Some grid cells contain multiple stations located within the 0.05° x 0.05° area. If more than one station was located within a single grid cell, an average of the station measurements was used for comparison against the modeled SM estimates. Therefore, the total number of grid cells suitable for evaluation equaled 78 based on a total of 101 stations.

### 4.1.1   Timeseries evaluation

Figure 3 presents the OL, DA-CDF (i.e., CDF-matched), and DA-NoCDF (i.e., no CDF-matching) estimated SM timeseries and their comparison with the in-situ measurements at two grid cells from two different networks. These example sites were selected because they reflect the performance of SM assimilation across two different climate zones. The Ngari network test site (Figs. 3(a) and 3(b)) represents a cold and arid climate while the the Maqu network test site (Figs. 3(c) and 3(d)) is located in a cold and humid climate.

For the Ngari network test site, MERRA2 forced modeled estimates overestimate the SM for all model simulations, Fig. 3(a). For MERRA2, the DA-NoCDF run has the lowest RMSE while the DA-CDF run shows the highest RMSE. In addition, DA-NoCDF captures the measured values within the mean $\pm$ standard deviation ($\mu \pm \sigma$) range after 8 September 2016 while the CDF-matched SMAP retrievals move the DA value in the opposite direction to the in-situ measurements. MERRA2-forced simulations exhibit improved consistency as the SM magnitude decreases with the approaching winter months. IMERG exhibits much better temporal consistency with the measurements throughout the study period shown in Fig. 3(b). For IMERG, the DA-CDF run has the lowest RMSE while the DA-NoCDF run has the highest RMSE. However, even the largest RMSE difference

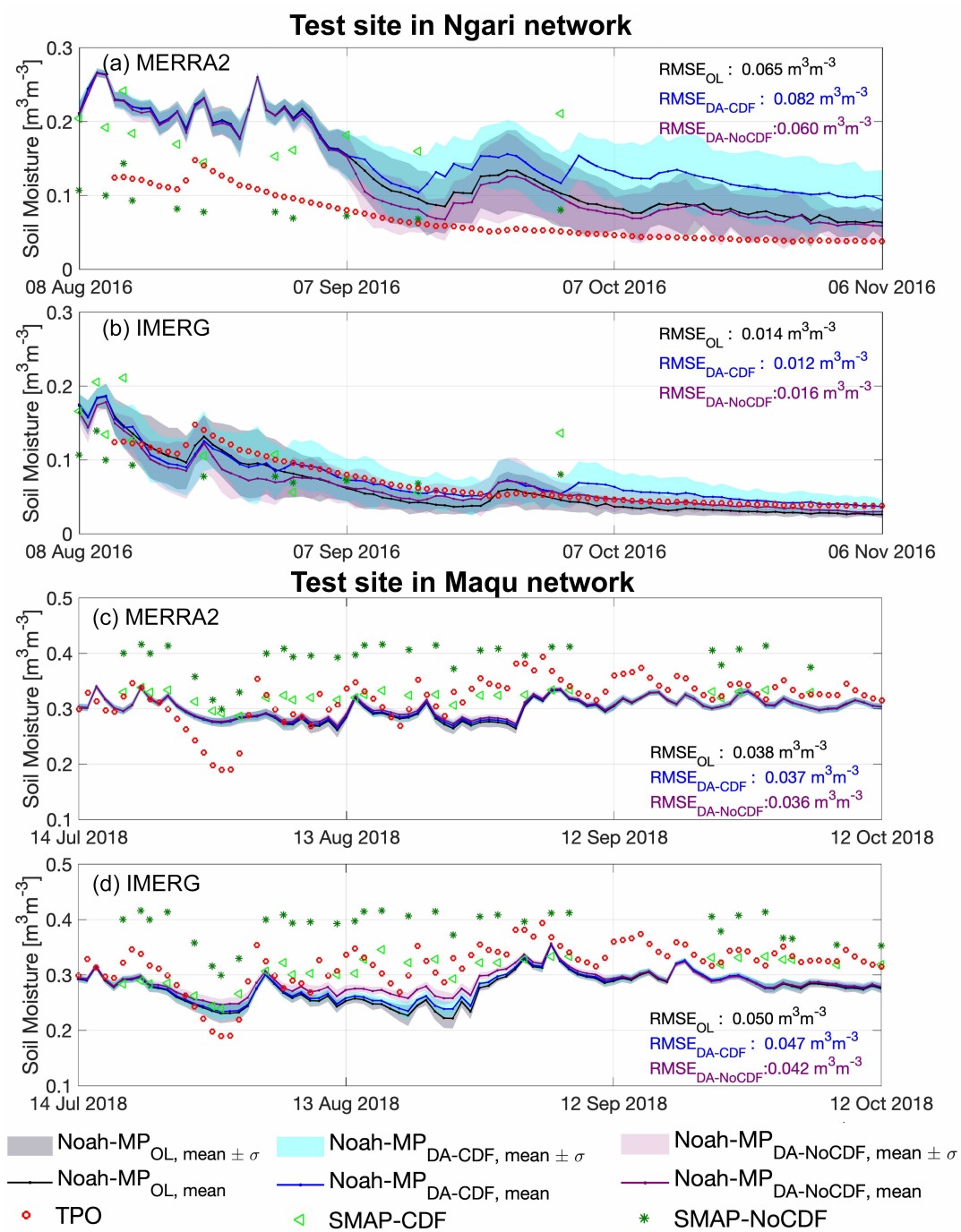

**Figure 3.** Comparative timeseries of open loop (OL) and data assimilation (DA) estimated surface (top 5 cm) soil moisture. The solid line represents the ensemble mean whereas the shaded areas represent ± 1 standard deviation ($\sigma$) across the full ensemble. DA-CDF= assimilation with CDF-matching; DA-NoCDF= assimilation without CDF-matching; TPO= Tibetan Plateau Observatory measurements; SMAP-CDF= SMAP retrieval value after CDF-matching; SMAP-NoCDF= original SMAP retrieval value.

(DA-CDF versus DA-NoCDF) is less than 0.004 m$^3$ m$^{-3}$ indicating the limited influence of assimilation at this location. The modeled values show localized underestimation as well as overestimation during different periods in the timeseries. For the cold and arid test site, IMERG exhibits lower RMSE as compared to the MERRA2 boundary condition estimates, Figs. 3(a) and 3(b).

Figures 3(c) and 3(d) present the Maqu test site timeseries for the MERRA2 forced and IMERG forced simulation runs, respectively. The MERRA2 runs display better temporal consistency with the measurements as compared to the IMERG runs. In Fig. 3(c), the DA-NoCDF run exhibits the lowest RMSE while the OL run has the highest RMSE magnitude. However, the differences between the RMSE magnitudes for the different MERRA2 runs are minimal (i.e., less than 0.002 m$^3$ m$^{-3}$). In Fig. 3(d), for IMERG the lowest RMSE is computed for the DA-NoCDF run while the OL has the highest RMSE magnitude. However, the difference in the RMSE magnitudes is higher than the values in Fig. 3(c). There is a negative bias (underestimation) apparent in all the IMERG runs after 1 August 2018. For the cold and humid test site, MERRA2 displays better performance as compared to the IMERG boundary condition estimates, Figs. 3(c) and 3(d).

Figure 3 shows the presence of biases in the modeled estimates and SMAP SM retrievals with respect to the in-situ measurements. Relative to MERRA2, IMERG-based SM estimates have lower RMSE for the sample location in the Ngari network and higher RMSE for the location in the Maqu network. This indicates the importance of precipitation boundary conditions in terms of SM estimation across locations of varying climatology (i.e., arid versus humid). The magnitude of state update for DA-NoCDF is generally larger than DA-CDF. However, the magnitude of the update increments is limited by the model parameters (such as wilting point and maximum SM capacity) and model and retrieval error assumptions (via ensemble uncertainty).

### 4.1.2 Statistical analysis

Relevant statistics were computed using all the measurements (from all the networks) available from October 2015 to September 2020 in conjunction with the corresponding Noah-MP modeled estimates. Table 2 presents mean bias, RMSE, unbiased RMSE, and correlation (R) computed for the OL, DA-CDF, and DA-NoCDF estimated SM. The individual statistics were calculated for each grid cell separately and were then averaged to represent the domain-averaged statistical performance of the modeled SM. The total number of grid cells used for comparison is equal to 78. A majority of the Ngari, Naqu, and CTP-SMTMN network stations are situated at locations where SMAP L3 retrievals have limited availability (Figs. 1(a) and 6(f)). These high elevation locations are completely frozen or partially frozen during a considerable part of the year leading to limitations in the applicability of the tau-omega algorithm used to retrieve soil moisture information from the SMAP observed

**Table 2.** Statistics of OL and DA soil moisture estimates (2015 to 2020) computed with respect to the soil moisture measurements across the Tibetan Plateau. All values are in units of m$^3$ m$^{-3}$ unless otherwise indicated. Mean refers to the average of all the stations included within the network. OL = Open Loop, DA-CDF = CDF-matched SMAP retrieval assimilation, and DA-NoCDF = data assimilated estimates without CDF matching of the SMAP retrievals.

| Tibetan Plateau | MERRA2 | | | IMERG | | |
| --- | --- | --- | --- | --- | --- | --- |
| Statistic | OL | DA-CDF | DA-NoCDF | OL | DA-CDF | DA-NoCDF |
| Mean bias | 0.070 | 0.070 | 0.059 | 0.031 | 0.033 | 0.025 |
| Confidence interval$_{95\%}$ limits- bias | 0.012 | 0.012 | 0.012 | 0.011 | 0.011 | 0.011 |
| Mean RMSE | 0.130 | 0.130 | 0.122 | 0.106 | 0.106 | 0.100 |
| Confidence interval$_{95\%}$ limits- RMSE | 0.007 | 0.007 | 0.008 | 0.008 | 0.008 | 0.008 |
| Mean unbiased RMSE | 0.066 | 0.066 | 0.060 | 0.066 | 0.064 | 0.061 |
| Confidence interval$_{95\%}$ limits- unbiased RMSE | 0.004 | 0.004 | 0.003 | 0.004 | 0.004 | 0.004 |
| Median relative RMSE [-] | 1.873 | 1.873 | 1.794 | 1.507 | 1.507 | 1.480 |
| Mean R | 0.295 | 0.300 | 0.370 | 0.327 | 0.321 | 0.447 |

brightness temperatures (O'Neill et al., 2014). Given that little or no assimilation occurs over several stations, several of the statistics computed for the OL, DA-CDF, and DA-NoCDF estimated soil moisture are quite similar.

MERRA2 and IMERG exhibit similar relative results, i.e., the lowest mean bias, RMSE, unbiased RMSE, and relative RMSE is computed for the DA-NoCDF run. Similarly, the highest correlation is also observed for the DA-NoCDF run. For MERRA2, in terms of mean bias the OL/DA-CDF and DA-NoCDF intersect at the 95% confidence interval limit (0.070±0.012

m$^3$ m$^{-3}$ versus 0.059±0.012 m$^3$ m$^{-3}$). Similar values are computed for RMSE (0.130±0.007 m$^3$ m$^{-3}$ versus 0.122±0.008 m$^3$ m$^{-3}$) and unbiased RMSE (0.066±0.004 m$^3$ m$^{-3}$ versus 0.060±0.003 m$^3$ m$^{-3}$). The IMERG mean bias, RMSE, and unbiased RMSE, however, overlap within the 95% confidence interval limits (columns 5-7 in Table 2).

Relative RMSE is calculated as the ratio of the RMSE to the standard deviation of the state variable (SM). The median relative RMSE highlights the relative accuracy of the majority of the grid cells. A relative RMSE of less than 0.7 indicates medium

or high goodness-of-fit depending on the state variable (McCuen, 2016). In terms of comparative values, the DA-NoCDF runs for both MERRA2 and IMERG show lower median relative RMSE magnitudes than the OL and DA-CDF estimates. Overall, it is observed that the IMERG statistical values are lower than the corresponding MERRA2 values, thereby indicating better performance of the IMERG-forced model estimates as compared to MERRA2 across the Tibetan Plateau.

### 4.2 Spatial analysis of OL versus DA

Figure 4 shows the difference in spatial patterns of the SM estimated by the OL and the DA-CDF/DA-NoCDF simulations during the summer (April to September) and winter (October to March) months. This temporal grouping is motivated by the precipitation climatology (i.e., summer monsoon versus winter westerlies) of the region (Dhar and Nandargi, 2003), which also

influences the irrigation patterns in the region. Two main crop seasons are noted across South Asia, i.e., the summer (Kharif) crop and the winter (Rabi) crop (Biemans et al., 2016). Precipitation, snowmelt, and ground water extraction are the main sources of river runoff that provides water for irrigation (Armstrong et al., 2018).

The magnitudes of DA minus OL values shown in the summer maps are relatively small compared to the magnitudes in the winter maps for all DA experiments. This feature suggests that there is a relatively higher consistency between the OL and DA-CDF/DA-NoCDF runs (i.e., smaller DA minus OL magnitudes) during the summer months when the bulk of the precipitation occurs, especially in the lower latitudes, as compared to the winter months. A spatial feature apparent in Figs. 4(b), 4(d), 4(f), and 4(h) is the occurrence of large differences in areas surrounding the major rivers in the lower latitudes ($\lesssim 31°$ N). The location of these large differences indicates the influence of irrigation on the water budget. Fig. 6(c) shows the map of the total percentage of irrigated area per grid cell that corresponds well with the cropland landcover type shown in Fig. 2(d). These three maps highlight the increase in model estimated SM by the assimilation of raw (i.e., no CDF matching applied) SMAP retrievals in the irrigated cropland grid cells. Further, comparing the MERRA2 maps (Figs. 4(a), 4(b), 4(e), and 4(f)) with the IMERG maps (Figs. 4(c), 4(d), 4(g), and 4(h)) it appears that the influence of the boundary conditions used (MERRA2 versus IMERG) is damped by more dominant influencing factors such as anthropogenic irrigation and seasonal precipitation. In other words, similar spatial patterns in DA minus OL are visible in both the MERRA2 and IMERG forced model estimates.

The DA-NoCDF simulation exhibits higher differences with the OL relative to the DA-CDF simulation. Therefore, Fig. 5 and Fig. C1 were generated to further dissect the spatial patterns in these differences with respect to landcover and soil texture. Fig. 5 presents the OL and MERRA2-forced DA-NoCDF joint PDFs (shown here as fractions of total landcover type grid cells) for the winter months of the 2016 water year. The bar graph in subplot 5(h) provides the percentage of grid cells for each landcover type that have at least one instance of SMAP retrieval assimilation. The highest percentage is observed for grid cells belonging to the cropland type.

Linear regression coefficients included in all the subplots of Fig. 5 represent the slope between the two axes. If the slope is >1 then, in general, the variable on the y-axis (here DA-NoCDF) has greater soil moisture magnitudes than the x-axis (here OL). Forest (subplot 5(a)), savannas (subplot 5(c)), and cropland (subplot 5(e)) landcover types show >1 linear regression coefficients, indicating that, in general, the SMAP assimilation increases the soil moisture magnitude across grid cells belonging to these landcover types. Interesting to note is that the percentage of grid cells with assimilation is quite different for these three landcover types (forest=10%, savannas=40%, and cropland=80%). It is difficult to ascertain the exact cause of the generally higher soil moisture magnitudes for the DA-NoCDF estimates relative to the OL for pixels included in savannas due to the small sample size. Approximately 1.4% of the total grid cells included in the study domain belong to the land cover type savan-

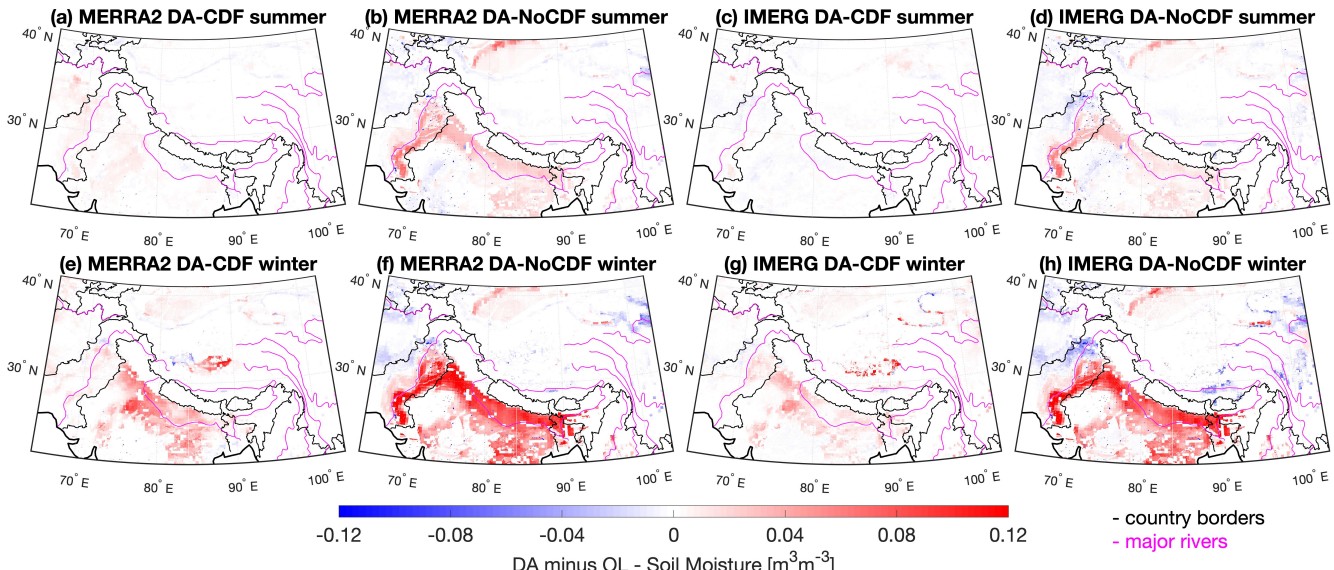

**Figure 4.** Differences between the mean soil moisture estimated by the OL and DA simulations during the summer (April 2016 to September 2016) versus the winter months (October 2015 to March 2016) highlight: 1) the unmodeled irrigation signal across croplands, and 2) the relatively higher influence of assimilation on soil moisture estimates during the winter period as compared to the summer period. DA-CDF= assimilation of CDF-matched SMAP retrievals; DA-NoCDF= no CDF-matching of SMAP retrievals.

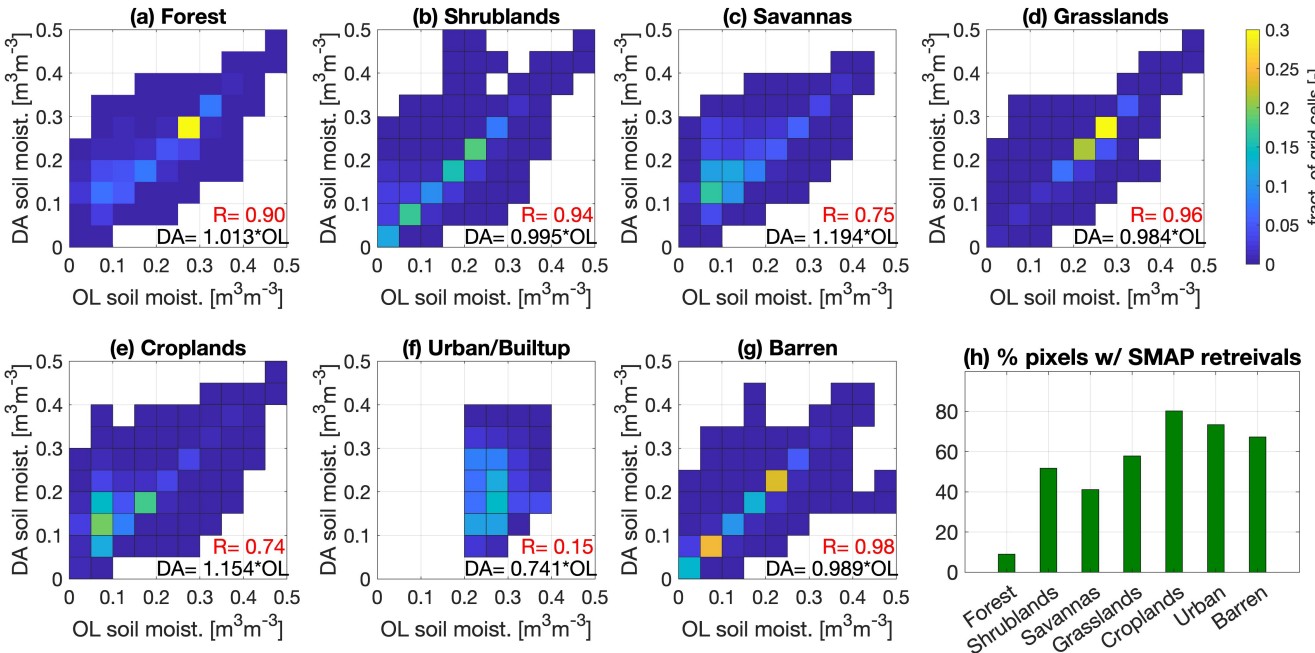

**Figure 5.** Comparison of OL versus DA-NoCDF estimated soil moisture according to the dominant landcover types present within the study domain. The OL and DA-NoCDF joint PDFs (presented here as fractions of grid cells) are computed from the LIS runs with MERRA2 boundary conditions during the winter months of WY 2016.

nas of which only 40% of the pixels have SMAP retrievals available for assimilation. For shrublands (subplot 5(b)), grasslands (subplot 5(d)), urban/built-up (subplot 5(f)), and barren (subplot 5(g)) landcover types, the linear regression coefficients are <1 indicating that, in general, the SMAP assimilation decreases the soil moisture magnitude across grid cells belonging to these landcover types. The lowest regression coefficient is computed for the urban/built-up landcover type.

The correlation coefficients for savannas, croplands and urban/built-up are $\leq 0.75$ and are lower than for the other landcover types, which suggests that SMAP SM assimilation alters the SM estimates across grid cells belonging to these three landcover types the most (Note: if the SM assimilation caused no change, the OL and DA SM estimates would be nearly identical, and hence the correlation coefficient between the two would equal 1.). The lowest correlation is computed for the urban/built-up landcover type, of which 70% of the grid cells underwent assimilation, however, this landcover type only represents 0.4% of the total domain grid cells (Table A1). Similar results were observed for the IMERG-forced simulation as well (results not shown). The OL and MERRA2-forced DA-NoCDF joint PDFs categorized with respect to soil texture types did not yield any distinctive patterns and are included in Appendix C for reference.

## 4.3 Irrigation impact

The unavailability of in-situ measurements across different land cover types limits a direct validation of the DA-CDF and DA-NoCDF estimated soil moisture across the lower part of the study domain. The influence of irrigation is analyzed through an indirect approach using the GMIA maps of irrigated areas. In South Asia, irrigation is implemented through routing of the: i) river runoff (contributed by snowmelt and precipitation), ii) discharge from storage reservoirs such as dams, and iii) water pumped from subsurface aquifers, using a network of canals and tube wells (Chambers, 1988). The GMIA total irrigation-equipped area map in Fig. 6(e) visualizes this practice as high magnitudes are observed in the areas surrounding the major rivers in Pakistan, India, and Bangladesh.

Irrigation is not explicitly modeled in the Noah-MP land surface modeling environment. Therefore, to investigate the effect of SM assimilation on irrigated areas in further detail, the maps of temporal mean normalized innovation (NI) were compared against the GMIA total irrigation-equipped area map. NI (detail in Sect. 3.3.3) represents the difference between the observations (i.e., SMAP SM retrievals) and the modeled *a priori* estimates. A positive NI value indicates that the *a priori* state estimate is less than the observed value while a negative NI value indicates that the *a priori* state estimate is greater than the observed value. For an unbiased, linear, optimal assimilation framework, the NI sequence exhibits a mean of 0 and a standard deviation equal to 1 over time. Therefore, high positive or negative NI values reveal the presence of bias either in the model estimates or the assimilated retrievals.

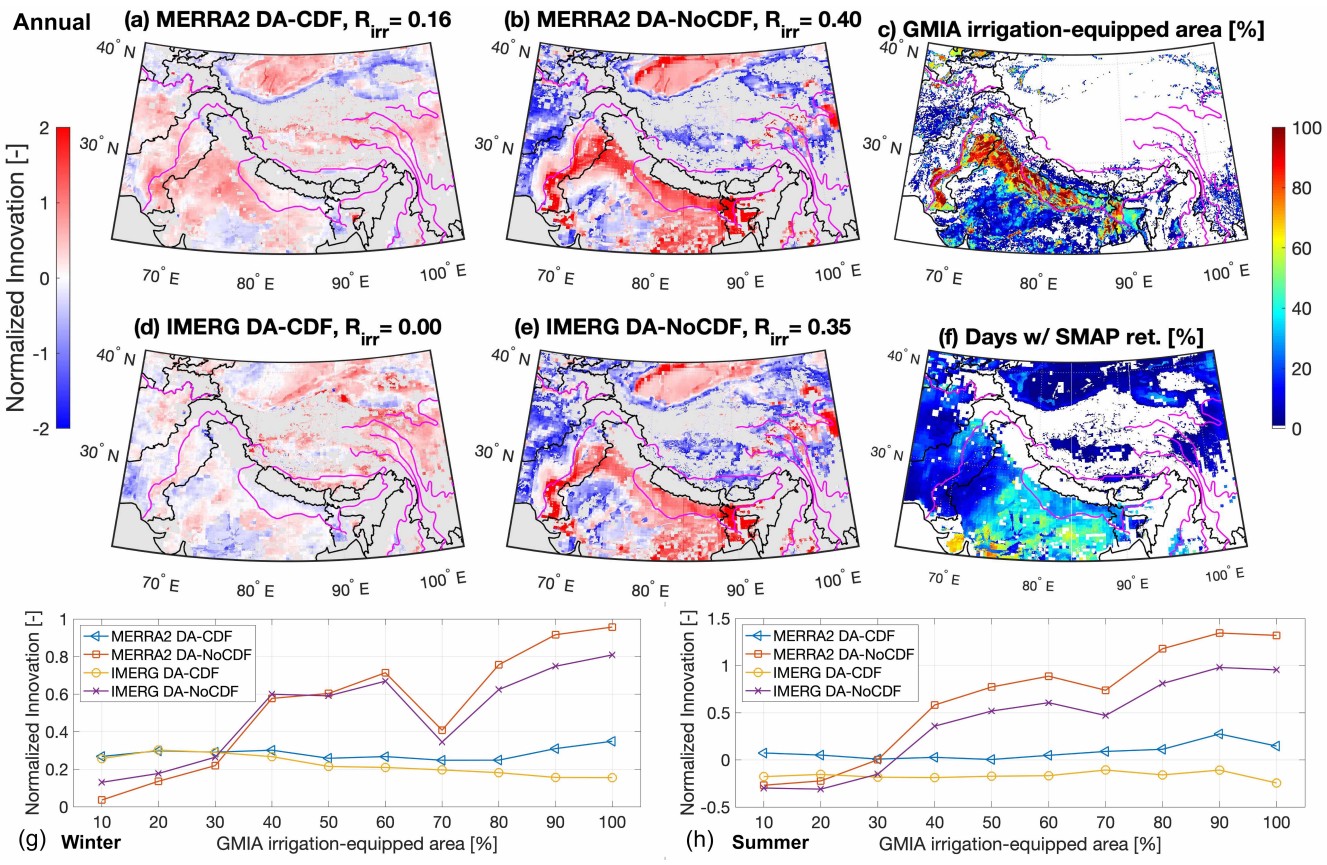

**Figure 6.** The spatial patterns in normalized innovation (NI) maps present the results of assimilating CDF-matched (DA-CDF) versus raw SMAP SM retrievals (DA-NoCDF), subplots (a), (b), (d), and (e). Grey areas represent grid cells where no assimilation occurred due to missing SMAP SM retrievals. The improved spatial correlation with respect to irrigation-equipped area ($R_{irr}$) for both of the DA-NoCDF maps (subplots (c) and (d)) highlights the correction of SM biases due to an unmodeled hydrologic process, i.e., irrigation. Subplots (g) and (h) underscore the increase in NI magnitude for both DA-NoCDF (MERRA2 and IMERG) simulations as the total irrigation-equipped area increases for summer and winter months, respectively. Subplot (c) presents the total percentage of irrigated area per grid cell developed from the Global Map of Irrigated Areas (GMIA) dataset provided by the Food and Agriculture Organization. Subplot (f) shows the % of total days in the study period on which SMAP retrievals were assimilated.

A number of distinct features can be observed in the NI maps presented in Fig. 6. MERRA2 DA-CDF and DA-NoCDF, and IMERG DA-NoCDF spatial patterns show positive NI values in Pakistan (Indus Basin) and the areas surrounding the Ganges River in India, Figs. 6(a), 6(b), and 6(e). Comparing the location of these positive NIs with the GMIA total irrigation-equipped area map (Fig. 6(c)), it is apparent that the SMAP retrievals have higher SM magnitudes across irrigated areas. SMAP retrievals implicitly contain the effects of irrigation and subsequently transfer that information to the modeled estimates via assimilation.

Hence, the water budget across these locations was corrected as information related to an unmodeled soil moisture source was effectively incorporated into the land surface model. Figures 6(g) and 6(h) show the general increase in mean NI magnitudes during the winter and summer months, respectively, as the percentage of irrigation-equipped area increases. NIs computed from the MERRA2 and IMERG DA-CDF runs, however, do not display this pattern.

     Further comparing the MERRA2 and IMERG DA-NoCDF NI maps with the water storage trends identified by Fig. 1 in

Girotto et al. (2017) and Fig. 2 in Loomis et al. (2019), the locations in the northwestern part of India that show negative water storage trends (resulting from groundwater pumping for purposes of irrigation) are spatially consistent with high positive NI values. The additional water introduced into the hydrologic cycle via pumping from subsurface aquifers is captured by the SMAP SM retrievals and is then used to condition the modeled estimates via assimilation.

     The spatial patterns in NI show different magnitudes (and even different signs) at some locations for DA-CDF versus DA-

385 NoCDF. The visible difference in NI signs is due to the implementation of CDF matching of the assimilated retrievals during the DA-CDF simulation. If the model estimates are biased, traditional data assimilation generally does not result in optimal estimates (Zhang and Moore, 2015). Mapping the observation CDF to a biased model CDF would ultimately transfer the model bias into the CDF-matched observations. Therefore, in cases where the model estimates are inherently biased, assimilation of CDF-matched retrievals could update the *a priori* state estimates in the wrong direction. This phenomenon is apparent in

IMERG DA-CDF versus IMERG DA-NoCDF NI maps across the irrigated areas and the Tibetan Plateau.

     One interesting pattern to note is the presence of highly negative NI values across the high elevation areas (Hindukush mountains) in the western part of the domain in the DA-NoCDF maps (subplots 6(b) and 6(e)). Comparing the DA-NoCDF NI maps with the DA minus OL map in Fig. 4, it is apparent that the high NI values did not manifest into high DA minus OL values. A high NI magnitude does not necessarily lead to a subsequently high update. If the model state error variance is quite

low, the denominator in Eq. 3 will be a small value that can then result in a large NI if the nominator (innovation) is relatively large. However, a low model state error variance results in a reduced Kalman gain (due to $C_{y_t z_t}$), and hence, the computed update will be relatively small.

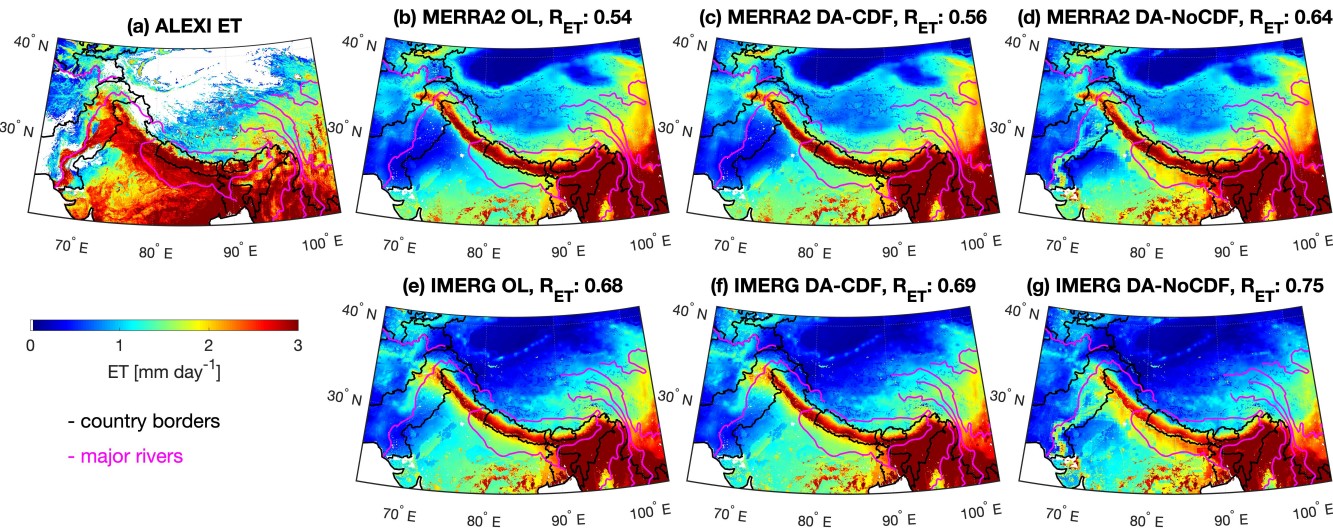

**Figure 7.** Comparative maps of modeled evapotranspiration (ET) with respect to the ALEXI evapotranspiration estimates (Sect. 3.2.3) for WY 2016. DA-NoCDF maps show relatively higher spatial consistency with ALEXI ET, particularly in areas surrounding the major rivers in lower latitudes ($<30°$). The correlation values ($R_{ET}$) indicate the spatial consistency between annual mean ET estimated by ALEXI and the corresponding Noah-MP simulation.

High NI magnitudes are observed in the Indus Basin even though assimilation occurred during $<20\%$ of the total days (in the study period) at these locations. This suggests that the quantitative effect of SMAP SM retrieval assimilation is not primarily based on the assimilation frequency, but rather the large differences between the SMAP and *a prioi* estimates. The DA-CDF versus DA-NoCDF results seen here are similar to the experiments conducted by Kumar et al. (2015) to evaluate SM retrievals across irrigated areas. Their study showed that bias correction of observations via CDF matching can lead to the removal of the information pertaining to the unmodeled processes from the observations when the estimation bias stems from the absence of such processes in the model.

## 4.4 Influence on water and carbon cycle

SM is an important component of the water cycle. It is, therefore, expected that changes in the SM estimates would translate into changes in hydrologic variables that are dependent on SM such as evapotranspiration (ET). ET is composed of evaporation from the soil and vegetation as well as transpiration from the vegetation. While ET is used to represent the water cycle in this section, gross primary production (GPP) and solar-induced chlorophyll fluorescence (SIF) are utilized as vegetation proxies that represent the carbon cycle.

In order to diagnose the influence of SMAP SM assimilation on ET, the mean annual ET from the MERRA2 and IMERG-forced OL, DA-CDF, DA-NoCDF simulations is analyzed. Figure 7 highlights the improved spatial consistency (relative to ALEXI ET) of the DA-NoCDF estimates (subplots 7(d) and 7(g)) compared to the OL (subplots 7(b) and 7(e)) and DA-CDF ET (subplots 7(c) and 7(f)). The spatial correlation of mean annual ET calculated with respect to the ALEXI ET for the MERRA2 runs increases from 0.54 for the OL to 0.56 for DA-CDF and 0.64 for the DA-NoCDF estimates. Similarly, there is an increase in the spatial correlation of the IMERG runs from 0.68 for the OL to 0.69 and 0.75 for the DA-CDF and DA-NoCDF estimates, respectively. The DA-NoCDF estimates for both sets of boundary conditions show relatively higher spatial correlation with the ALEXI ET, particularly in the Indus River Basin, where surface irrigation is significant. All three of the MERRA2 estimates show higher ET magnitudes across the Tibetan Plateau as compared to the IMERG runs, which corresponds well with the higher positive bias computed in the MERRA2-forced SM estimates (see Table 2). All of the IMERG simulations exhibit better overall spatial correlation with ALEXI ET relative to the MERRA2 runs.

Comparing the spatial patterns in ET magnitudes with the GMIA irrigation-equipped area map (Fig. 6(c)), it can be seen that the mean ET magnitudes across irrigated areas, particularly across the Indus basin, increased for DA-NoCDF simulations (Figs. 7(d) and 7(g)) relative to the OL. However, this feature is absent in the DA-CDF simulations (Figs. 7(c) and 7(f)). The spatial patterns observed in the DA minus OL SM (see Figs. 4(f) and 4(h)) are similarly shown in the ET maps (Figs. 7(d) and 7(g)) in terms of higher ET magnitudes observed for grid cells belonging to the cropland landcover type.

Further investigation of this feature highlighted the correction of SM and ET in irrigated areas via SMAP assimilation. It is expected that as the irrigation percentage increases the surface SM would also increase. The increase in SM, in general, translates into an increase in ET. Figure 8 shows the increase in ALEXI ET as the percentage of irrigated area (Fig. 6(c)) in each grid cell increases. In contrast, the OL and DA-CDF estimates do not capture this behavior, and alternatively, show declining ET values for regions with 40% or more total irrigation-equipped area when using the MERRA2 boundary conditions. The IMERG OL and DA-CDF estimates show approximately the same decreasing trend. However, the DA-NoCDF estimates corrected the decreasing magnitudes for grid cells with >40% total irrigation-equipped area for both sets of precipitation boundary conditions.

The ALEXI ET dataset serves as an independent evaluation source for OL, DA-CDF, and DA-NoCDF ET estimates. The ET magnitudes for all the modeled runs are lower than the ALEXI ET, which could be attributed to the absence of relevant processes (e.g., surface irrigation) in Noah-MP, whereas the ALEXI product implicitly includes this information. Although ALEXI is a modeled dataset, it is based on remote sensing data and has been shown to detect irrigation (Knipper et al., 2019).

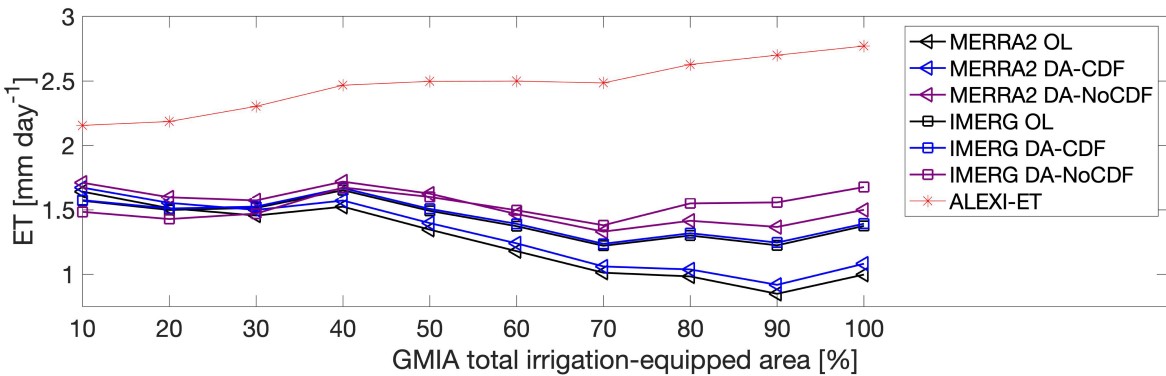

**Figure 8.** The magnitude of average evapotranspiration (ET) increased as the percentage of irrigated area within the grid cell increased.

These results suggest assimilation of SMAP SM retrievals in the absence of CDF-matching can help correct for some of the
missing physics in the Noah-MP land surface model.

Figures 9(a), (b), and (c) were created to further dissect the influence of SMAP assimilation on the water and carbon cycle
over irrigated regions. The test site selected contains 88% total irrigation-equipped area and belongs to the cropland landcover
type. Noah-MP divides the soil profile into four layers. Figure 9(a) shows the monthly temporal variation in near-surface (first
soil layer, L1) and root-zone (second soil layer, L2) soil moisture at this location. The first (top) soil layer (L1) is 5 cm deep,
while the second layer (L2) extends 35 cm below that. L1 estimates for all simulations exhibit a seasonal variation in the
surface SM with the major peak occurring in Feb and a secondary peak in Aug. The DA-NoCDF runs for both sets of boundary
conditions depict a higher seasonal amplitude as compared to the OL. Comparing the L1 values with the L2 values, the damping
of the seasonal variation amplitude is apparent in L2, i.e., the influence of assimilation on surface SM is not proportionally
translated into the root-zone SM. However, compared to the OL, the DA-NoCDF estimates for L2 do exhibit seasonal variation
(albeit to a limited extent). The DA-CDF estimates were quite similar to the OL L1 and L2 estimates and are thus excluded
from the graph for visual clarity. Figure 9(b) highlights the translation of L1 SM temporal patterns into ET estimates. ALEXI
ET displays much higher magnitudes of ET throughout the year. The DA-NoCDF simulations exhibit better consistency with
ALEXI ET as compared to the OL and DA-CDF ET for both sets of precipitation boundary conditions.

Figure 9(c) presents the impact of SM on vegetation in terms of gross primary production (GPP) and solar-induced fluo-
rescence (SIF). Compared to the FluxSat GPP (Sect. 3.2.4), the magnitude of OL and DA (Noah-MP) GPP observed at this
location is relatively small. However, similar seasonal variability (not magnitude) is observed in all the Noah-MP simulations
similar to the FluxSat GPP (peaks in Feb/Mar and Aug/Sep). The OL, DA-CDF (not shown in figure), and DA-NoCDF GPP
estimates exhibit high similarity and do not differ significantly throughout the year. A possible explanation for this behaviour

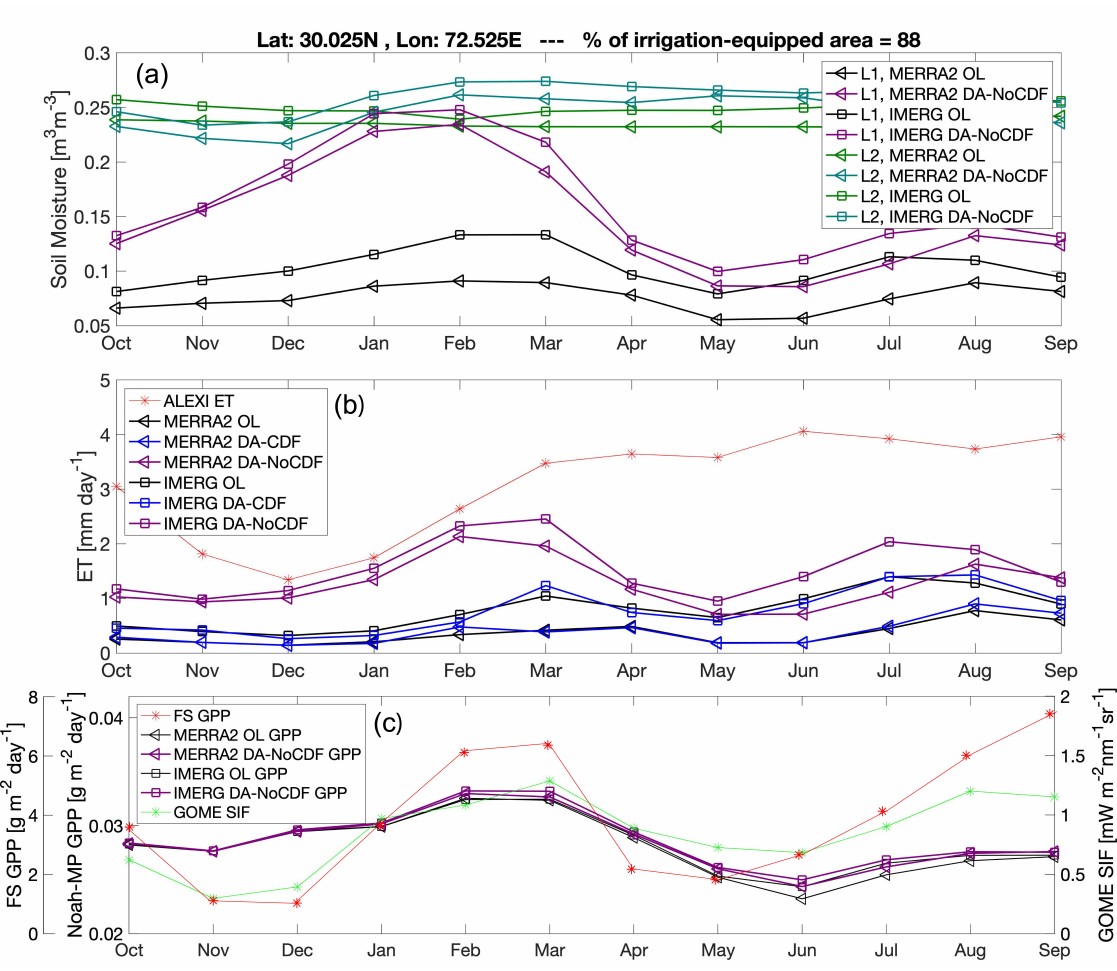

**Figure 9.** Influence of SMAP soil moisture (SM) assimilation on an irrigated location is assessed through soil moisture of successive soil layers (L1 and L2), evapotranspiration (ET) and the corresponding behavior of the dynamic vegetation. ALEXI ET (Sect. 3.2.3), FluxSat gross primary production (FS GPP; Sect. 3.2.4), and GOME solar-induced chlorophyll fluorescence (SIF; Sect. 3.2.5) are used as evaluation datasets. (a) L1 = layer 1 near-surface SM and L2 = layer 2 root-zone SM. Noah-MP modeled ET exhibits similar temporal patterns as the near-surface SM (L1); however, root-zone (L2) SM and GPP are not correspondingly modulated. DA-CDF= assimilation with CDF-matching; DA-NoCDF= assimilation without CDF-matching.

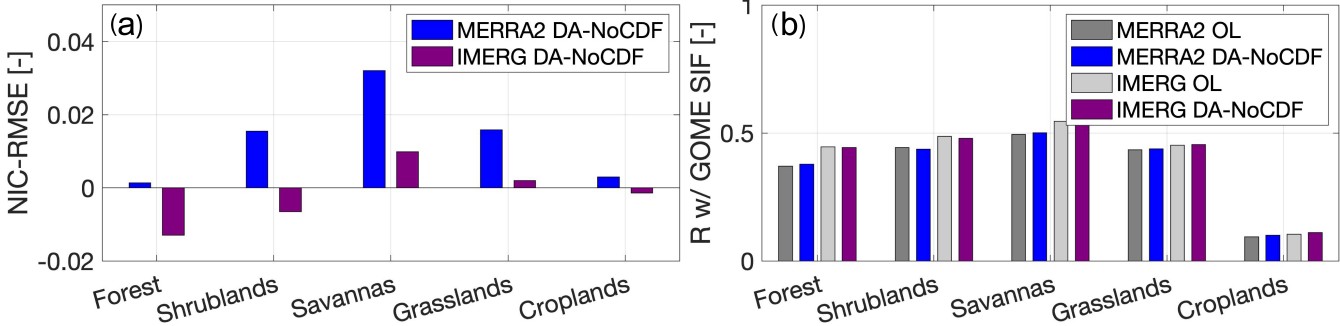

**Figure 10.** (a) Normalized information content (NIC) with respect to RMSE ($\text{RMSE}_{OL}$ versus $\text{RMSE}_{DA-NoCDF}$) is computed through comparison with FluxSat gross primary production (FS GPP). (b) Correlation with GOME solar-induced chlorophyll fluorescence (SIF) depicts the spatiotemporal consistency between the Noah-MP modeled GPP and GOME SIF. Data from the summer months of water years 2016-2019 were used to compute the metrics.

is that vegetation transpiration is more dependent on root-zone SM than surface SM. In Fig. 9(b), it is seen that the change

in near-surface (L1) SM is largely modulated in terms of root-zone (L2) SM. In general, root-zone SM tends to maintain low variation throughout the year. Thus, it is expected that assimilation of surface SM retrievals may not significantly impact the dynamic vegetation.

FluxSat GPP and Noah-MP GPP were compared with respect to dominant landcover types and it was observed that the SMAP assimilation did not influence the vegetation within any of the landcover type grid cells to a high extent, Fig. 10.

Even the highest percent improvement in the RMSE, computed for savannas (normalized information content (NIC) = 4.5%, see Appendix D for formula) during the summer months was <5%. The correlations between GOME-SIF and the different Noah-MP modeled estimates are similar in magnitude and do not highlight any significant influence of SMAP assimilation (OL versus DA-NoCDF) with respect to individual landcover types. Comparing these results to the vegetation optical depth (VOD) assimilation implemented by Kumar et al. (2020), it seems that the modeled GPP estimates are more improved by

assimilating VOD than surface SM. In the context of land surface modeling with Noah-MP, surface SM exhibits a weaker influence on GPP as compared to VOD. This is because SM has an indirect effect on GPP, whereas assimilation of VOD has a direct impact on plant biomass, and hence, on GPP. Kumar et al. (2020) found that SM had a higher control over ET and GPP during moisture-limited conditions.

## 5   Discussion

Statistics included in Table 2 show the relatively better performance of DA-NoCDF estimates as compared to the OL and DA-CDF estimates via evaluation with in-situ soil moisture measurements across the Tibetan Plateau. Direct comparison of SMAP

soil moisture retrievals with in-situ measurements yielded higher relative RMSE than all other estimates (Table S1). SMAP soil moisture retrievals also had the lowest mean bias and RMSE. However, only 30 grid cells were available for comparison with in-situ measurements as SMAP data has extensive gaps across the Tibetan Plateau due to frozen soil conditions. SMAP

retrievals are provided on a 36 km EASE Grid and contain frequent data gaps in space and time. The Noah-MP model was run at a relatively fine resolution of 0.05° (∼5 km) and provides continuous data without any spatiotemporal gaps (along with lower relative RMSE values). Therefore, while SMAP retrievals contain important information, the Noah-MP model estimates provide a more consistent dataset without spatiotemporal gaps associated with frozen soil conditions, swath width limitations, or radio frequency interference.

The Noah-MP simulation results in Sec. 4 highlight that CDF-matching removes the irrigation signal from the SMAP soil moisture retrievals, and therefore, better results are obtained across croplands for simulations without any CDF-matching. Optimal data assimilation is based on the assumption that the forward model and the observed data are unbiased, which is one motivating factor for conducting CDF-matching of retrievals. Considering the current study domain, it is apparent that the forward model unbiasedness assumptions are violated across irrigated areas. Hence, mapping the retrieval climatology to a

biased land surface model climatology is not a viable bias correction approach for satellite-based retrievals. The spatial patterns in the DA-CDF estimated soil moisture across irrigated areas (Fig. 4) highlight this issue.

In an effort to comply with the unbiased forward model assumptions in the EnKF assimilation algorithm, assimilation using an anomaly-based approach (i.e., one that is zero mean by construct) was also tested. In this approach, the retrieval mean was mapped to the land surface model mean and updates were computed using the resultant anomalies. Anomaly-based

assimilation results (Fig. S1) showed that for heavily irrigated areas assimilation estimates closely mimic the OL estimated soil moisture throughout the year whereas DA-NoCDF is able to update the soil moisture based on the information in the SMAP observations, particularly during the winter months. In terms of general spatial patterns (Fig. S2), the anomaly-based assimilation results were similar to the DA-CDF soil moisture estimates such that relatively higher soil moisture values were found across some irrigated areas during the winter. Further details regarding the anomaly-based assimilation experiment are

included in the supplement document, see Sect. S2.

It is important to note that while in the presented study, estimation accuracy is better for assimilation without CDF-matching, the results might be different for other cases. That is, the assimilation of retrievals without bias adjustment may not improve the estimation accuracy as compared to CDF-matched satellite retrievals. In this particular study, the SMAP soil moisture retrievals are able to effectively capture the irrigation signal, and as such, help improve the Noah-MP modeled soil moisture estimates

via assimilation. However, there is the possibility that assimilation of a different soil moisture retrieval product may degrade

the accuracy of the modeled estimates depending on the inherent biases in that given soil moisture retrieval. It is important that the model physics be improved as well so that the regional hydrologic processes are accounted for, resulting in a more representative model which could then be used for bias correction of satellite retrievals.

Irrigation is primarily carried out via manually operated canals, open channels, and ground pumping across South Asia. The amount of water contributed by irrigation in South Asian croplands changes in magnitude during different seasons, however, it remains non-negligible over the course of the entire year (Biemans et al., 2016). Therefore, assumptions regarding higher contribution of irrigation to the regional water cycle during winter and negligible contribution during the summer months are not appropriate. Hence, implementation of CDF-matching only during certain months would have limitations in this region. That is, there is a need to develop an irrigation module that would be able to represent the regional irrigation practices, and therefore, properly account for the contribution of water transported via manually operated irrigation schemes in the local water balance.

The results in Sect. 3.2.4 highlight the limitations in information transfer from updated surface soil moisture to root-zone soil moisture or to the vegetation. Compared to root-zone soil moisture, the influence of SMAP soil moisture assimilation was greater on surface soil moisture. One potential method of transferring surface soil moisture information to deeper soil layers could entail the development of a soil modeling routine that has higher hydrologic coupling between the individual soil layers. However, an important point to consider is that with an increase in the hydrologic coupling between surface and deep soil layers, the complexity of the land surface model would also increase as new parameters are required to model the feedback loop between adjacent soil layers. Similarly, information transfer between the updated surface soil moisture and the vegetation states is also limited.

## 6   Conclusions

Soil moisture estimation across South Asia was implemented in this study by assimilating Soil Moisture Active Passive (SMAP) soil moisture retrievals into a land surface model. The Noah-MP land surface model was run within the NASA Land Information System software framework to simulate the regional land surface processes. Precipitation boundary conditions (in different experiments) were provided by the NASA Modern-Era Retrospective Analysis for Research and Applications (MERRA2) and GPM Integrated Multi-satellite Retrievals (IMERG) products. SMAP retrieval assimilation was implemented using two approaches: i) DA-CDF= bias correction of observations prior to assimilation using CDF-matching, and ii) DA-NoCDF = SMAP retrieval assimilation without CDF-matching. CDF-matching of the observations to the modeled estimates was applied in an effort to correct the distribution moments of the SMAP soil moisture retrievals.

Comparison of assimilated and model-only soil moisture estimates against in-situ measurements showed the relative improvement in soil moisture by assimilating SMAP retrievals. The IMERG DA-NoCDF simulation exhibited the best goodness-of-fit and reduced the mean bias and RMSE by 8.4 and 9.4% across the Tibetan Plateau. The results presented in Sect. 4 highlight that SMAP soil moisture assimilation decreased the magnitude of error (Table 2), and suggest improvements in the spatiotemporal soil moisture patterns (Figs. 3 and 6) and associated evapotranspiration (Fig. 7), particularly over irrigated areas. However, the influence on evapotranspiration did not proportionally translate into changes in the carbon flux.

An important feature of SMAP retrieval assimilation observed in this study is the suggested correction of state estimation biases resulting from missing physics in the land surface model (unmodeled hydrologic process), i.e., irrigation. Information about the exact quantity and timing of irrigation practices is generally not publicly available except for a few parts of the globe. Simulating the complex regional irrigation scheme is a difficult task that is further complicated by the inaccessibility of relevant pumping data, manual operation of reservoirs, and unsystematic canal to field irrigation. The framework described in this paper could potentially be used to infer information regarding irrigation patterns and practices using an inverse method. Brocca et al. (2018) used coarse-scaled soil moisture retrievals to quantify the amount of water used for irrigation. A similar methodology can be explored that uses the difference between the OL and DA estimated soil moisture across croplands to infer information regarding the water quantity supplied by irrigation.

Considering the lack of in-situ observations available for use in this study, it is difficult to clearly ascertain the influence of assimilation without CDF-matching in areas that are not irrigated. Across the Tibetan Plateau, DA-NoCDF estimates exhibit the lowest RMSE. However, the evaluation of DA-NoCDF estimates across unirrigated areas in the southern part of the study domain is limited by the scarcity of ground data. A follow-on study should explore the influence of including (as well as excluding) CDF-matching in areas that are not irrigated. This experiment could help explore suitable approaches for incorporating the information obtained from satellite retrievals to correct the modeled estimates without introducing additional bias to the modelled estimates. In a broader perspective, there is a need to develop a bias correction technique for satellite retrievals that is independent of the accuracy or bias of the model climatology. Using in-situ measurements for pre-processing of the satellite retrievals would be one potential method. Current efforts in South Asia by various governmental and non-governmental organizations to measure in-situ soil moisture would benefit the development of suitable methods of bias correction of satellite observations.

The utility of L-band radiometry for soil moisture estimation is limited by the soil emission depth associated with passive microwave (∼5 cm) and the data gaps in the soil moisture retrievals. These data gaps are due to the presence of snow, ice, frozen soil, dense vegetation, radio frequency interference instances, and swath width limitations. The influence of SMAP

soil moisture retrieval assimilation was primarily limited to surface soil moisture, compared to root-zone soil moisture, across locations where SMAP soil moisture retrievals were available for assimilation. One method of transferring surface soil moisture information to deeper soil layers could entail the development of a soil modeling routine that has higher hydrologic coupling between the individual soil layers. While it may improve the information transfer to deeper soil layers, the complexity of the land surface model would also increase considerably with the addition of new parameters that would better control the feedback between adjacent soil layers.

Improvements in the fine-scale spatial and temporal patterns in soil moisture were observed even though the retrievals being assimilated were at a much coarser scale than the model grid (36 km versus 0.05°). These results highlight the potential applicability of the described framework for regions where measured data are scarce as well as where accurate and consistent soil moisture estimates do not currently exist. A follow-on study to be explored based on the results of the described experiments is the routing of streamflow using modeled runoff to analyze the effect of soil moisture assimilation on runoff and river discharge. Antecedent soil moisture conditions affect the soil permeability and infiltration capacity. Therefore, it is expected that improvements in soil moisture estimation could translate into improved streamflow estimates.

**Appendix A:  Soil texture and landcover across study domain**

Table A1 presents the predominant soil texture and landcover classes and their respective percentages across the study domain shown in Fig. 1.

**Appendix B:  Meteorological forcing perturbations**

Table B1 includes the meteorological forcing perturbation used for the OL and DA simulations Kwon et al. (2019).

**Appendix C:  OL versus DA estimates with respect to soil texture**

Figure C1 displays the OL and MERRA2-forced DA-NoCDF joint PDFs (shown here as fractions of total grid cells) categorized with respect to the soil texture types for the winter months of the 2016 water year. The bar graph in subplot C1(h) provides the percentage of grid cells belonging to each soil texture type that have at least one instance of SMAP retrieval assimilation. The soil types that included sand or loam exhibited regression coefficients >1 (except for loamy sand). Grid cells belonging to loamy sand (subplot C1(b)) , silty clay (subplot C1(h)), and clay (subplot C1(i)) soil types exhibited regression coefficients 1, indicating a general decrease in SM magnitude after SMAP assimilation. However, the regression coefficients of all three of

**Table A1.** List of soil texture and landcover classes (and their respective percentages) found within the study domain presented in Fig. 1.

| | Soil texture | | | Landcover | |
|---|---|---|---|---|---|
| **Class** | **no. of grid cells** | **% of total grid cells** | **Class** | **no. of grid cells** | **% of total grid cells** |
| Sand | 12528 | 4.04 | Forest | 43669 | 14.1 |
| Loamy Sand | 322 | 0.10 | Shrublands | 62654 | 20.2 |
| Sandy Loam | 18753 | 6.05 | Savannas | 4244 | 1.4 |
| Silt Loam | 2098 | 0.68 | Grasslands | 41306 | 13.3 |
| Loam | 188716 | 60.91 | Croplands | 67366 | 21.7 |
| Sandy Clay Loam | 14132 | 4.56 | Urban/ Built-up | 1269 | 0.4 |
| Clay Loam | 28885 | 9.32 | Snow/Ice | 1027 | 0.3 |
| Silty Clay | 35 | 0.01 | Barren/Sparsely vegetated | 78338 | 25.3 |
| Clay | 23048 | 7.44 | Ocean | 9952 | 3.2 |
| Water | 10805 | 3.49 | | | |
| Other (ice/lakes/water bodies) | 10503 | 3.39 | | | |

**Table B1.** Perturbation parameters applied to meteorological forcing fields for both the open loop and data assimilation simulations. M= multiplicative; A= additive.

| **Perturbed meteorological forcing** | **Perturbation type** | **Standard deviation** | **Cross-correlations with perturbations** | | | |
|---|---|---|---|---|---|---|
| | | | P | SW | LW | Tair |
| Precipitation (P) | M | 0.5 | - | -0.8 | 0.5 | -0.1 |
| Shortwave radiation (SW) | M | 0.3 | -0.8 | – | -0.5 | 0.3 |
| Longwave radiation (LW) | A | 50 W m$^{-2}$ | 0.5 | -0.5 | - | 0.6 |
| Near-surface air temperature (Tair) | A | 1 K | -0.1 | 0.3 | 0.6 | - |

these soil texture types are close to one, and therefore, do not reinforce any significant influence of SMAP assimilation on grid cells belonging to these particular soil texture types.

**Appendix D: Statistical metrics**

The following formulas were used to calculate the relevant statistics described in Sect. 4:

$$\text{Bias} = \sum_{t=1}^{T} (y_s - y_m) \tag{D1}$$

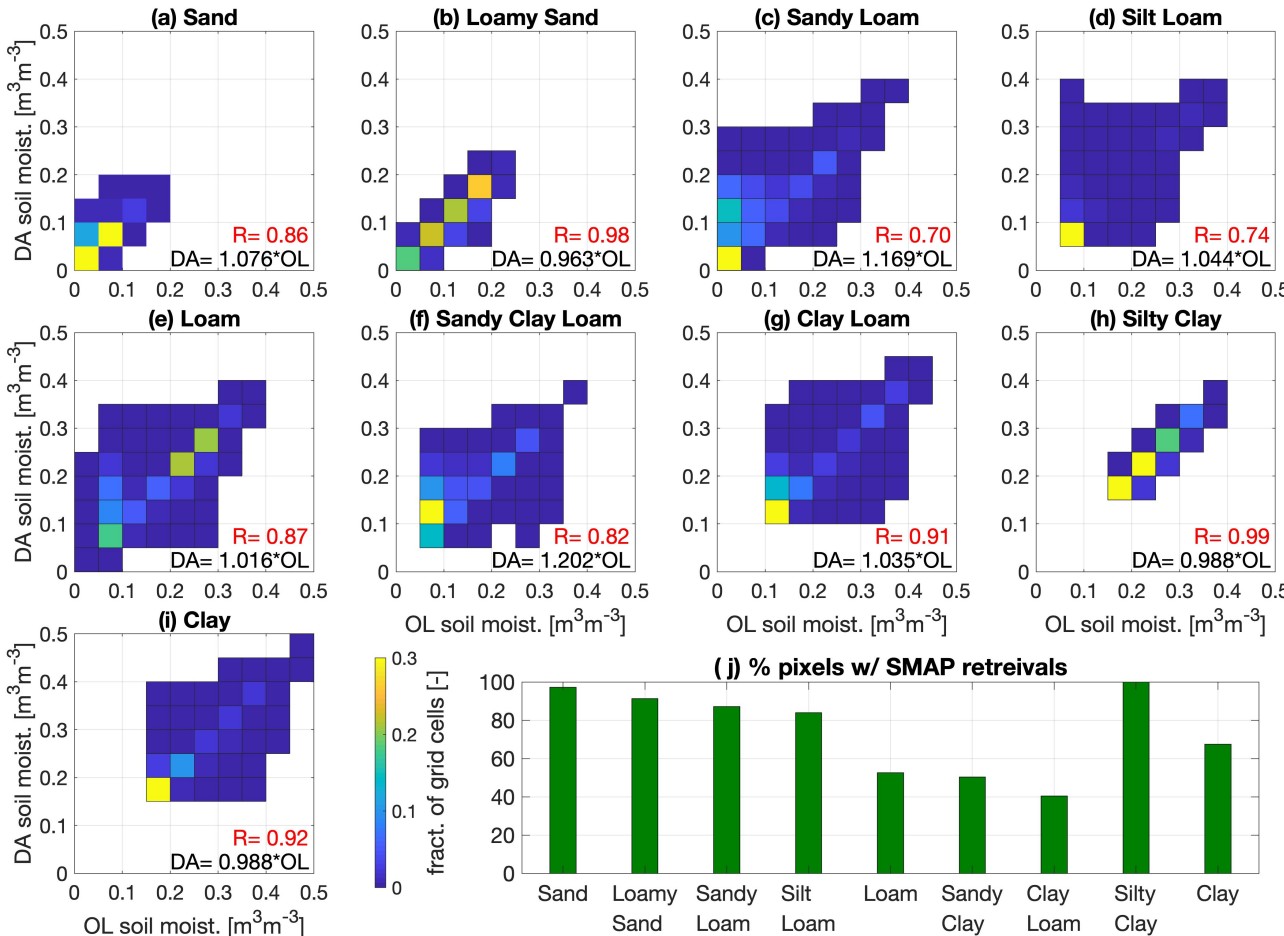

**Figure C1.** Comparison of OL versus DA-NoCDF estimated soil moisture according to the dominant soil texture types present within the study domain. The OL and DA-NoCDF joint PDFs (presented here as fractions of grid cells) are computed from the LIS runs with MERRA2 boundary conditions during the winter months of WY 2016.

$$\text{RMSE} = \sqrt{\frac{\sum_{t=1}^{T}(y_s - y_m)^2}{T}} \tag{D2}$$

$$\text{Unbiased RMSE} = \sqrt{\frac{\sum_{t=1}^{T}((y_s - \overline{(y_s - y_m)}) - y_m)^2}{T}} \tag{D3}$$

$$\text{Relative RMSE} = \frac{\text{RMSE}}{\sigma_{y_s}} \tag{D4}$$

$$\text{Confidence interval}_{95\%} \text{ limits} = \pm 1.96 * \frac{\sigma_X}{\sqrt{N}} \tag{D5}$$

where $y_s$ equals the ensemble mean of the OL/DA-CDF/DA-NoCDF soil moisture estimate, $y_m$ is the in-situ soil moisture measurement, $\sigma_{y_s}$ is the standard deviation of the ensemble mean soil moisture over time, $T$ is the total number of data instances in time at a given location in space, $X$ is the array containing bias/RMSE values computed for each comparative grid cell, and $N$ is the total number of (in-situ measurements versus modeled estimates) comparative grid cells. The overbar represents temporally averaged values. The cross-correlation, R, between variables x and y was computed as:

$$R = \frac{\sum_{t=1}^{T}(x - \bar{x})(y - \bar{y})}{\sqrt{\sum_{t=1}^{T}(x - \bar{x})^2 \sum_{t=1}^{T}(y - \bar{y})^2}} \tag{D6}$$

The fractional normalized information content, $\text{NIC}_{RMSE}$, improved in terms of RMSE due to assimilation was computed as:

$$\text{NIC}_{RMSE} = \frac{\text{RMSE}_{OL} - \text{RMSE}_{DA}}{\text{RMSE}_{OL}} \tag{D7}$$

where $\text{RMSE}_{OL}$ is the root mean squared error (RMSE) for the Open Loop and $\text{RMSE}_{DA}$ is the RMSE for the DA-CDF or DA-NoCDF experiment.

**Acronyms and abbreviations**

| | |
|---|---|
| ALEXI | Atmosphere-Land Exchange Inverse |
| CDF | Cumulative distribution function |
| DA | Data assimilation |
| DA-CDF | Data assimilation with CDF matching |
| DA-NoCDF | Data assimilation without CDF matching |
| EnKF | Ensemble Kalman filter |
| ET | Evapotranspiration |
| GMIA | Global Map of Irrigation Areas |
| GOME-2 | Global Ozone Monitoring Experiment 2 |
| GPP | Gross primary production |
| IMERG | Integrated Multi-satellite Retrievals for Global Precipitation Measurement |
| L1 | Layer 1 near-surface soil moisture |
| L2 | Layer 2 root-zone soil moisture |
| LIS | Land Information System |
| MODIS | Moderate Resolution Imaging Spectroradiometer |
| MERRA2 | Modern-Era Retrospective analysis for Research and Applications |
| NI | Normalized innovation |
| SIF | Solar-induced fluorescence |
| SM | Soil moisture |
| SMAP | Soil Moisture Active Passive |
| OL | Open loop |
| VOD | Vegetation optical depth |

*Code and data availability.* The NASA Land Information System source code was downloaded from https://github.com/NASA-LIS/LISF. SMAP soil moisture retrievals were downloaded from https://nsidc.org/data/SPL3SMP/. Soil moisture measurements across the Tibetan Plateau are available at https://ismn.earth/en/. FluxSAT Gross Primary Production is available at https://avdc.gsfc.nasa.gov/pub/tmp/FluxSat_GPP/, while the ALEXI evapotranspiration dataset can be accessed at https://lpdaac.usgs.gov/products/eco3etalexiv001/. GOME-2 Fluo-

rescence dataset can be downloaded from https://avdc.gsfc.nasa.gov/pub/data/satellite/MetOp/GOME_F/. Noah-MP modeled soil moisture estimates analyzed in the paper can be accessed at http://hdl.handle.net/1903/28297.

*Author contributions.*  J.A.A. conducted the described experiments, contributed to the development of the study methodology, and compiled

the manuscript; B.A.F. served as Co-Investigator to the project, contributed to the development of the study methodology, and helped edit the manuscript; S.V.K. served as the Principle Investigator to the project, developed the NASA Land Information System software platform used to run the assimilation experiments, and contributed to the selection and acquiring of evaluation datasets.

*Competing interests.*  The authors declare that there are no competing interests present.

*Acknowledgements.*  The authors would like to extend their gratitude to Dr. Yonghwan Kwon and Dr. David Mocko for providing guidance

in setting up the NASA LIS assimilation framework. Additional thanks are extended to Dr. Christopher Hain and Dr. Martha Anderson for sharing the ALEXI dataset. Funding for this study was provided by the NASA Understanding Changes in High Mountain Asia contract (80NSSC2OK1531), while the high-performance computing facility at the University of Maryland provided the super-computing platform to run the assimilation experiments.

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
