# Peer review of "Soil moisture estimation in South Asia via assimilation of SMAP retrievals"

_Hydrology and Earth System Sciences, 2021_

## Author Comment (AC1)

Reviewer# 2

Thank you for your comments. We have endeavored to reply to each and every point that was raised by the reviewer. The referee comments are written in black with the author replies directly below in red.

Major point

Comment: The comparison with in situ soil moisture measurements is the only direct and independent evaluation. However, the stations are not located in the regions with higher irrigation-equipped areas. These are the regions with higher impact, and with some indirect evaluation via the signal on evapotranspiration. While the results are consistent, the main conclusions are not fully supported by independent evaluation. Therefore, I suggest that this limitation is highlighted in the conclusions and abstract. For example, Line 473: "and improved the spatiotemporal soil moisture patterns (Figs. 3 and 7) and associated evapotranspiration (Fig. 8), particularly over irrigated areas." Since there are no direct observations of evapotranspiration and Soil moisture over the irrigated areas, I would recommend to say that the results "suggest improvements".

Response: Thank you for your comment. The evaluation is indeed limited by the unavailability of ground measurements across the irrigated areas in South Asia. To highlight this point, the following textual modifications have been implemented:

Lines 9 to 17

"Across the Tibetan Plateau, DA-NoCDF reduced the mean bias and RMSE by 8.4% and 9.4%, even though assimilation only occurred during less than 10% of the study period due to frozen (or partially frozen) soil conditions. The best goodness-of-fit statistics were achieved for the IMERG DA-NoCDF soil moisture experiment. The general lack of publicly available in-situ measurements across irrigated areas limited a domain-wide direct model validation. However, comparison with regional irrigation patterns suggested correction of biases associated with an unmodeled hydrologic phenomenon (i.e., anthropogenic influence via irrigation) as a result of SMAP soil moisture retrieval assimilation. The greatest sensitivity via assimilation was observed in cropland areas. Improvements in soil moisture potentially translate into improved spatiotemporal patterns of modeled evapotranspiration, although limited influence from soil moisture assimilation was observed on modeled processes within the carbon cycle such as gross primary production."

Lines 472 to 473

"The results presented in Sect. 4 highlight that SMAP soil moisture assimilation decreased the magnitude of error (Table 2), and suggest improvements in the spatiotemporal soil moisture patterns (Figs. 3 and 7) and associated evapotranspiration (Fig. 8), particularly over irrigated areas."

Line 476 to 477

"An important feature of SMAP retrieval assimilation observed in this study is the suggested correction of state estimation biases resulting from missing physics in the land surface model (unmodeled hydrologic process), i.e., irrigation."

Comment: Also on this point, the title can be a bit misleading since there is no clear evidence of improved soil moisture across irrigated areas, but we see the impact of the assimilation across irrigated areas. Therefore, I also suggest a change in the title to clearly reflect the manuscript content.
Response: We have changed the title to be more reflective of the paper to:
"Soil moisture estimation across South Asia via SMAP retrieval assimilation"

Minor details
Comment: line 185 " perturbations used by Kwon et al. (2019) (Table 2)" For completeness, I would recommend replicating Table 2 of Kwon 2019 (if there were no changes?) in the appendix for example.
Response: Thank you for the suggestion. The following table has been included in the appendix:

Table B1. Perturbation parameters applied to meteorological forcing fields for both the open loop and data assimilation simulations. M= multiplicative; A= additive.

| Perturbed meteorological forcing | Perturbation type | Standard deviation | Cross-correlations with perturbations | | | |
|---|---|---|---|---|---|---|
| | | | P | SW | LW | Tair |
| Precipitation (P) | M | 0.5 | - | -0.8 | 0.5 | -0.1 |
| Shortwave radiation (SW) | M | 0.3 | -0.8 | – | -0.5 | 0.3 |
| Longwave radiation (LW) | A | 50 W m$^{-2}$ | 0.5 | -0.5 | - | 0.6 |
| Near surface air temperature (Tair) | A | 1 K | -0.1 | 0.3 | 0.6 | - |

Comment: Figure 3: Please define acronyms in figure captions. E.g. "TPO" in red symbols are the observations ? SMAP-CDF are the colocated SMAP observations after CDF matching ? Also in Figure 10, e.g. panel c) FS GPP == FluxSat GPP ?
Response: Thank you for the recommendation. We have clarified the acronyms within the figure captions.

"Figure 3: Comparative timeseries of open loop (OL) and data assimilation (DA) estimated surface (top 5 cm) soil moisture. The solid line represents the ensemble mean whereas the shaded areas represent +/- 1 standard deviation ($\sigma$) across the full ensemble. DA-CDF= assimilation with CDF-matching; DA-NoCDF= assimilation without CDF-matching; TPO= Tibetan Plateau Observatory measurements; SMAP-CDF= SMAP retrieval value after CDF-matching; SMAP-NoCDF= original SMAP retrieval value."

"Figure 10: Influence of SMAP soil moisture (SM) assimilation on an irrigated location is assessed through soil moisture of successive soil layers (L1 and L2), evapotranspiration (ET) and the corresponding behavior of the dynamic vegetation. ALEXI ET (Sect. 3.2.3), FluxSat gross primary production (FS GPP; Sect. 3.2.4), and GOME solar-induced chlorophyll fluorescence (SIF; Sect. 3.2.5) are used as evaluation datasets. (a) L1 = layer 1 near-surface SM and L2 = layer 2 root-zone SM. Noah-MP modeled ET exhibits similar temporal patterns as the near-surface SM (L1); however, root-zone (L2) SM and GPP are not correspondingly modulated. DA-CDF= assimilation with CDF-matching; DA-NoCDF= assimilation without CDF-matching."

"Figure 11: (a) Normalized information content (NIC) with respect to RMSE ($RMSE_{OL}$ versus $RMSE_{DA-NoCDF}$) is computed through comparison with FluxSat gross primary production (FS GPP). (b) Correlation with GOME solar-induced chlorophyll fluorescence (SIF) depicts the spatiotemporal consistency between the Noah-MP modeled GPP and GOME SIF. Data from the summer months of water years 2016-2019 were used to compute the metrics."

Comment: Lines 336-354: It's not clear what's the motivation of Figure 6. To link some potential impact of the DA as a function of soil texture ? While it was clear in Figure 5 for the land cover, I would recommend removing Figure 6, and just mention in the text that no clear link was found between soil moisture in OL vs DA-NoCDF for different soil textures.

Response: Similar to land cover type, Figure 6 was intended to examine the updates in the spatial patterns due to assimilation with respect to soil texture. The objective was to analyze if any meaningful conclusions could be drawn based on the different soil types. As was shown in Figure 6, there were no distinctive patterns with respect to varying soil compositions. According to Reviewer #1's suggestion, and in an effort to focus the discussion within the main text, Figure 6 has been moved to the appendix with the following text added to line 342:

"The OL and MERRA2-forced DA-NoCDF joint PDFs categorized with respect to soil texture types did not yield any distinctive patterns and are included in Appendix C for reference."

Comment: Line 47: please define PMW
Response: We have replaced the acronym with the full term, i.e., passive microwave.

---

## Author Comment (AC2)

Reviewer #1

Thank you for your insightful comments. We have made our best effort to address each and every point that was raised by the reviewers. The referee comments are written in black with the author replies directly below in red.

1. MAJOR: The main problem with the paper is that the better results are obtained when SMAP soil moisture data are assimilated without the correction of the BIAS. Of course, the correction of the BIAS in regions in which we have irrigation, that is not modelled by the Noah-MP version used in this study, will provide wrong results. However, without the correction of the BIAS, i.e., no CDF-matching (or other methods), the assimilation method is not correct as well. Indeed, the assimilation should correct the random error, not the BIAS. The bias between modelled and satellite data is related to multiple causes, differences in spatial scale, differences in parameterization (e.g., wilting point and saturation), differences in land cover specification. A BIAS between modelled and satellite data is expected. For instance, a positive BIAS is observed also in the summer period (see Figure 4 and 7) or in areas not irrigated (Figure 4), and it cannot be easily explained. I am sure that in Pakistan and northern India the satellite soil moisture data can contribute to see the unmodelled irrigation and hence to contribute to improve land surface modelling. However, the approach used in the study should be corrected. E.g., the Noah-MP model with the irrigation module can be run. BIAS correction only during summer (i.e., in which irrigation is negligible) can be implemented (not month by month). These are just two suggestion to the authors.

Response: Thank you for your comment. The main point highlighted in this paper is that CDF-matching removes the irrigation signal from the retrievals, and therefore, we note that better results are obtained across croplands for simulations without any CDF-matching. Optimal data assimilation is based on the assumption that the forward model and the observed data are unbiased, which is one motivating factor for conducting CDF-matching. It is clearly apparent that in the current case these assumptions are violated in some cropland areas. With that said, assimilation without CDF-matching provides the best results across irrigated areas and the Tibetan Plateau.

The suggestion of applying bias correction during summer only would not yield the desired results. The amount of water contributed by irrigation changes in magnitude during different seasons (higher during some months depending on the crop), however, it remains non-negligible throughout the year (Biemans et al., 2016). Figure 5 in Biemans et al. (2016) highlights the year-round contribution of irrigation to the regional water balance in the Indus Basin. We cannot assume that during the summer period there is no contribution of irrigation. Therefore, CDF-matching during some months and no CDF-matching during other months will not help in mitigating all the biases from the SMAP retrievals.

As an alternative, assimilation using an anomaly-based approach (i.e., one that is zero-mean by construct) was also tested. In the interest of limiting the scope of the study, these results were not added in the paper. In this approach, the retrieval mean was mapped to

the land surface model mean and updates were computed using the resultant anomalies such that:

$$\text{Observation anomaly } (s, t) = \text{SMAP soil moisture } (s, t) - \text{Mean SMAP moisture } (s)$$
$$\text{Observed value } (s, t) = \text{Mean NoahMP soil moisture } (s) + \text{Observation anomaly } (s, t)$$
$$s = \text{location in space}$$
$$t = \text{instance in time}$$

[Figure]

*Figure A: Comparative timeseries of OL and DA estimated surface (top 5 cm) soil moisture at an irrigation-equipped pixel. The solid line represents the ensemble mean whereas the shaded areas represent mean +/- standard deviation across the full ensemble. DA-CDF: anomaly-based assimilation; DA-NoCDF: no CDF-matching based assimilation.*

Figure A shows a sample timeseries for a location that is 80% irrigation-equipped. It is apparent that assimilation estimates (DA-CDF) after anomaly scaling closely mimic the OL estimated soil moisture throughout the year whereas DA-NoCDF is able to update the soil moisture based on the information in the SMAP observations, particularly during the winter months. This could be further explored if in-situ measurements were available. Unfortunately, there are no publicly available soil moisture datasets across the three primary river basins in South Asia, i.e., Indus, Ganges, and Brahmaputra, from 2015 onwards.

Figure B shows the differences between DA-NoCDF and DA-CDF estimated soil moisture values. The irrigation signal across the cropland regions is clearly seen. Apart from the croplands, negative differences are noted across much of the Tibetan Plateau. Comparison with in-situ measurements across the Tibetan Plateau revealed that the DA-NoCDF run had lower bias than the OL and DA-CDF runs. Using the model climatology to correct biases in the retrieval CDF does not take into account the model's own inherent biases. The validity of the assumption that the model climatology is not (or more or less) biased as compared to the satellite-based retrievals varies depending on the location specifications. Particularly, for areas such as South Asia where land surface modeling capabilities of contemporary LSMs need improvement in order to adequately represent

all the relevant land surface processes. As the reviewer noted, the source of the bias cannot be easily explained. However, the results included in the manuscript show that assimilation is potentially capable of correcting biases that originate from missing information in the land surface model, i.e., irrigation.

[Figure]

Figure B: Differences between the mean soil moisture estimated by the DA-NoCDF and DA-CDF simulations during the summer (April 2016 to September 2016) and the winter months (October 2015 to March 2016). Differences greater than 0.1 m$^3$/m$^3$ are, in general, observed across cropland regions. Negative values within the Tibetan Plateau are explained by the better accuracy of the DA-NoCDF relative to in-situ measurements. DA-CDF= assimilation of CDF-matched SMAP retrievals; DA-NoCDF= no CDF-matching of SMAP retrievals; GMIA= Global Map of Irrigated Areas.

The irrigation module included in Noah-MP was tested by a number of our colleagues who concluded that the simulation routine lacked adequate representation in terms of the regional irrigation patterns (Ghatak et al., 2018). Zaitchik et al. are currently working on

developing a more representative irrigation modeling routine for the study domain that would benefit future explorations of soil moisture across South Asia. Additionally, another follow-on study is planned to explore the influence of no CDF-matching on areas that are not irrigated, and how best to devise a suitable method of incorporating the information obtained from satellite retrievals to correct the modeled estimates without introducing additional bias to the modelled estimates. Some potential methodologies include Kornelsen and Coulibaly (2015), Lee et al. (2017), and Zhou and Grassotti (2020).

The framework used in this paper has some limitations. As the reviewer pointed out, a positive bias is observed across some pixels that have low irrigation-equipped area percentages. To acknowledge this limitation, the following text has been added to lines 489 to 494.

"Considering the lack of in-situ observations, it is difficult to ascertain the influence of assimilation without CDF-matching on areas that are not irrigated. Across the Tibetan Plateau, DA-NoCDF estimates exhibit the lowest RMSE. However, the evaluation of DA-NoCDF estimates across unirrigated areas in the southern part of the study domain is limited by the scarcity of ground data. A follow-on study would explore the influence of no CDF-matching on areas that are not irrigated and test suitable methods of incorporating the information obtained from satellite retrievals to correct the modeled estimates without introducing additional bias to the modelled estimates."

Biemans, H., Siderius, C., Mishra, A. and Ahmad, B., 2016. Crop-specific seasonal estimates of irrigation-water demand in South Asia. Hydrology and Earth System Sciences, 20(5), pp.1971-1982.

Kornelsen, K.C. and Coulibaly, P., 2015. Reducing multiplicative bias of satellite soil moisture retrievals. Remote Sensing of Environment, 165, pp.109-122.

Lee, J.H., Zhao, C. and Kerr, Y., 2017. Stochastic bias correction and uncertainty estimation of satellite-retrieved soil moisture products. Remote Sensing, 9(8), p.847.

Zhou, Y. and Grassotti, C., 2020. Development of a Machine Learning-Based Radiometric Bias Correction for NOAA's Microwave Integrated Retrieval System (MiRS). Remote Sensing, 12(19), p.3160.

2.  MODERATE: The title and the text are misleading. The study is carried out in a limited region between northern India and southern China and Pakistan, not South Asia. Moreover, from the results it is not shown that soil moisture is improved over irrigated areas, as the comparison with in situ data is carried out over non-irrigated sites. Please correct the title and the corresponding text.
Response: The paper has been retitled to:
    "Soil moisture estimation in South Asia via SMAP retrieval assimilation"

We use the term South Asia to identify the location of the study area. The three main river basins in South Asia, i.e., Indus, Ganges, and Brahmaputra, and the Tibetan Plateau are

the focus of this study. We have used the general and well-known term of South Asia to keep the title succinct and easily geographically identifiable.

One approach of evaluating the estimated soil moisture datasets was via comparison with in-situ measurements across the Tibetan Plateau. The secondary evaluation was indirect and based on known physical relationships. The Food and Agriculture Organization (FAO) provides a global map of irrigation-equipped areas (GMIA, Figure Ae). Section 4.3 is dedicated to the discussion of the soil moisture estimated by the OL, DA-CDF, and DA-NoCDF simulations. An analysis is carried out to connect the location of highly irrigated areas and their correspondence with the individual soil moisture datasets. Using rational deduction, we note that SMAP assimilation without CDF-matching is correcting the magnitude of soil moisture estimates across irrigated areas. As noted above, it would be ideal if in-situ observations in cropland areas were available. Unfortunately, there are no publicly available soil moisture datasets in the three main river basins of South Asia from 2015 onwards. In such a scenario, a viable option was to use rational deduction and spatial correlation with the GMIA irrigation data.

3.  MAJOR: The results of the comparison with in situ data are not robust. I believe that direct comparison of SMAP soil moisture against in situ observations provides a much better agreement with in situ data. Therefore, the data assimilation configuration is not optimal, and very likely the model error has been underestimated (or overestimated the SMAP observations error).

Response: Thank you for your comment.

Direct comparison of SMAP soil moisture retrievals with in-situ measurements yields higher relative RMSE than all other estimates. Only 30 grid cells are available for comparison with in-situ measurements as SMAP data has extensive gaps across the Tibetan Plateau due to frozen soil conditions. The following metrics were computed for the SMAP soil moisture retrievals and the modeled soil moisture estimates by comparing them with the relevant 30 grid cells containing in-situ data and available SMAP observations from 2015 onwards (see Table A). The SMAP soil moisture statistics are based on observations available on any day between 2015-2020, i.e., there are many temporal gaps in the SMAP soil moisture timeseries. However, the OL and DA-NoCDF statistics are computed from daily estimates. (Note: The statistics included in Table 2 of the manuscript take into account all 78 grid cells suitable for comparison with the modeled estimates (which do not have any data gaps).)

Table A: Statistical metrics computed with respect to in-situ measurements.

| Statistic | SMAP soil moisture | OL (MERRA2) | DA-NoCDF (MERRA2) | OL (IMERG) | DA-NoCDF (IMERG) |
|---|---|---|---|---|---|
| Mean bias $[m^3/m^3]$ | 0.02 | 0.07 | 0.06 | 0.03 | 0.02 |
| Mean RMSE $[m^3/m^3]$ | 0.10 | 0.13 | 0.12 | 0.11 | 0.10 |
| Mean relative RMSE [-] | 2.00 | 1.87 | 1.79 | 1.51 | 1.48 |
| Mean Correlation [-] | 0.48 | 0.30 | 0.37 | 0.33 | 0.45 |

SMAP soil moisture retrievals and DA-NoCDF (IMERG) yield the lowest mean bias and RMSE. However, the SMAP retrievals have the largest mean relative RMSE when compared against the in-situ measurements. Consistent with contemporary literature, the raw SMAP retrievals show the highest correlation with in-situ measurements. Important to note is the improvement in all statistics after assimilation as compared to the OL.

SMAP retrievals are provided on a 36 km grid and contain frequent data gaps in space and time. The Noah-MP model was run at a relatively fine resolution of 5 km and provides continuous data without any spatiotemporal gaps (along with lower relative RMSE values). Therefore, while SMAP retrievals contain important information, the Noah-MP model estimates provide a more consistent dataset without spatiotemporal gaps associated with frozen soil conditions, swath width limitations, or radio frequency interference.

A number of test experiments were carried out to ascertain the most representative model and SMAP soil moisture retrieval error values. In the interest of manuscript size limits, these results were not included in the paper. Model error standard deviation was increased from 0.02 $m^3/m^3$ to 0.1 $m^3/m^3$, while the SMAP error standard deviation was kept fixed at the standard value used in literature, i.e., 0.04 $m^3/m^3$. Similarly, the model error was fixed while the SMAP soil moisture error standard deviation was increased. Based on the test results, it was seen that the smallest bias and RMSE values were computed for model and SMAP soil moisture retrieval error standard deviations equal to 0.04 $m^3/m^3$. In order to clarify this point, the following text has been added at lines 226 to 231:

"Test simulations were conducted to ascertain the most suitable model and SMAP soil moisture retrieval error values (results not shown). Model error standard deviation was increased from 0.02 $m^3/m^3$ to 0.1 $m^3/m^3$, while the SMAP error standard deviation was kept fixed at the standard value used in literature, i.e., 0.04 $m^3/m^3$. Similarly, the model error was fixed while the SMAP soil moisture error standard deviation was increased. Based on the test results, it was seen that the smallest bias and RMSE values were computed for model and SMAP soil moisture retrieval error standard deviations equal to 0.04 $m^3/m^3$."

Also, I believe that the sample size for which the analysis has been carried is quite low, sample size should be added to assess the significance of the results.
We agree with the reviewer that the in-situ sample size is limited and would benefit from an increase in the number of measurement stations. With that said, we have worked diligently to obtain as many publicly-available datasets for model comparison as possible. As noted by the reviewer, an ideal evaluation strategy would entail the use of in-situ soil moisture measurements across all landcover types. However, there are no publicly available soil moisture measurements across the southern part of the study domain. That is, soil moisture measurements from 2015 onwards are not available for public use. Available in-situ measurements are limited to the Tibetan Plateau only.

On this basis, I believe such analysis should be improved and that meaningful inferences cannot be done due to, e.g., the very low correlation values or very high relative RMSE (>1.5 and should be lower than 0.7). Note that SMAP vs in situ provides R values greater

than 0.8 (see e.g., https://www.mdpi.com/2072-4292/10/4/535/htm, but there are several other papers). In my opinion, in the paper it is not shown that the assimilation improves soil moisture estimates, at least not a robust assessment, and not in irrigated areas (as in the title).

The paper shared above by the reviewer shows correlation between in-situ data and SMAP soil moisture retrievals across a differently defined study period (i.e., one year only from May 2015 to September 2016). Our study covers a longer time period (May 2015 to September 2020). Hence, the performance statistics are not directly comparable to one another. In addition, in the referenced paper, R>0.8 is for the Naqu network only. In this manuscript, in-situ measurements from the Naqu, Maqu, Ngari, and CTP-SMTMN networks are included in order to evaluate the performance of modeled datasets across a larger range of climates, and as a result, the correlation values are lower. Given that our study evaluates a larger period of time across a larger region of space, it is expected that the performance metrics would differ between the two different studies.

We have intentionally added the relative RMSE values to highlight that although assimilation is improving the estimates in the right direction, there is a need to further advance soil moisture estimation within this domain. The data-scarcity prevalent in the study area renders the development of comprehensive soil moisture products vital.

Since the traditional evaluation strategy of using in-situ measurements is limited across the study area, an indirect approach was used. The FAO provides a global map of irrigated areas (GMIA) that was used to infer information regarding the percentage of irrigation-equipped area within each grid cell. Figure 7 in the manuscript shows that as the percentage of irrigation equipped area increased, the normalized innovation, which contributes to the state update, also increased. Analogously, the spatial correlation for the DA-NoCDF simulation increased to 0.4 (IMERG forcings) and 0.35 (MERRA2 forcings) from 0.16 and 0.0 for the DA-CDF estimates, respectively. Since the OL and DA-CDF simulations are not properly considering the influence of irrigation on the soil moisture, physical rationality suggests that the DA-NoCDF estimates are updating the soil moisture in the right direction. Again, we agree with the reviewer that direct comparison against a robust in-situ network is always preferred, but for this particular study domain that is not an option. Hence, we made the best with the limited amount of information that was made available.

In order to address this framework limitation, the following text has been added to lines 356 to 358:
"The unavailability of in-situ measurements across different land cover types limits a direct validation of the DA-CDF and DA-NoCDF estimated soil moisture across the lower part of the study domain. The influence of irrigation is analyzed through an indirect approach using the GMIA maps of irrigated areas."

4. MODERATE: As expected, the impact of the assimilation on the evapotranspiration fluxes and on GPP is limited. It depends on the variable that is assimilated, i.e., surface soil moisture, but also on the coupling of such variable with root zone soil moisture and fluxes. It happened frequently that models are not able to transfer surface soil moisture information to deeper layer due to model and data assimilation technique

limitations. There's a lot of scientific literature on the topic. Please consider this important aspect in the assessment of the impact on fluxes, and possibly to improve the data assimilation framework employed in the study.

Response: Thank you for your comment. Figures 8 and 9 show that evapotranspiration (ET) benefits from assimilation (DA-NoCDF) as the magnitude of evapotranspiration is higher than the OL across irrigated areas. Also, the spatial correlation with the independent ALEXI dataset is also improved via assimilation. Figures 10 and 11 present the limited influence of assimilation on gross primary production (GPP). In the context of land surface modeling with Noah-MP, surface soil moisture exhibits a weaker influence on GPP as compared to ET. A first-order estimate of this influence is provided by the correlation between ET and GPP. The domain-wide spatiotemporal correlation between ET and GPP is equal to 0.654 and 0.650 while the correlation between vegetation transpiration and GPP is equal to 0.845 and 0.846 for the OL and DA-NoCDF simulations, respectively. The minute decrease in the correlation between ET and GPP after assimilation results from the decrease in the correlation between ET and vegetation transpiration after assimilation (0.582 versus 0.581). The magnitude of cross-correlation between ET and GPP is limited by the percentage of different landcover types within the study domain. The most dominant land cover type in the study domain is barren/sparse vegetation. Hence, the major influence of soil moisture assimilation across much of this domain manifests as surface evaporation rather than vegetation transpiration.

Figure 11 shows that the correlations between GOME-SIF and the different Noah-MP modeled estimates are similar in magnitude and do not highlight any significant influence of SMAP assimilation for major landcover types. In contrast, vegetation optical depth (VOD) assimilation implemented by Kumar et al. (2020) showed relatively higher improvement in modeled GPP estimates as compared to surface soil moisture assimilation. GPP is directly influenced by VOD via plant biomass, whereas the impact of surface soil moisture updates on GPP is indirect via rootzone soil moisture and ET. The following text included at lines 467 to 474 discusses these results.

"The correlations between GOME-SIF and the different Noah-MP modeled estimates are similar in magnitude and do not highlight any significant influence of SMAP assimilation (OL versus DA-NoCDF) with respect to individual landcover types. Comparing these results to the vegetation optical depth (VOD) assimilation implemented by Kumar et al. (2020), it seems that the modeled GPP estimates are relatively more improved by assimilating VOD than surface SM. In the context of land surface modeling with Noah-MP, surface SM exhibits a weaker influence on GPP as compared to VOD. This is because SM has an indirect effect on GPP, whereas assimilation of VOD has a direct impact on plant biomass, and hence, on GPP. Kumar et al. (2020) found that SM had a higher control over ET and GPP during moisture-limited conditions."

In order to transfer the soil moisture information to deeper soil layers, one possible future study could entail the development of a soil modeling routine that has higher hydrologic coupling between the individual soil layers. The following sentences have been added to lines 505 to 510:

"The influence of SMAP soil moisture retrieval assimilation was primarily limited to surface soil moisture, compared to root-zone soil moisture, across locations where SMAP soil moisture retrievals were available for assimilation. One method of transferring surface soil moisture information to deeper soil layers could entail the development of a soil modeling routine that has higher hydrologic coupling between the individual soil layers. While it may improve the information transfer to deeper soil layers, the complexity of the land surface model would also increase considerably with the addition of new parameters that would better control the feedback loop between adjacent soil layers."

SPECIFIC COMMENTS (P: page, L: line or lines)

5.  P1, Title: Please change by considering the general comment above.
Response: The title has been partially modified according to the referee's recommendation to:
"Soil moisture estimation across South Asia via SMAP retrieval assimilation"

6.  P3, L77: Typo "populace"
Response: Populace refers to people residing in an area. Thus, we used the term here to refer to all the people residing in the study area.

7.  P6, L128: Show in the map the location of in situ soil moisture stations with the name of the networks.
Response: Thank you for the comment. The network names have been added to Fig. 1.

[Figure]

8.  P7, L142: Please check the acronyms definition throughout the text, e.g., MODIS defined below, not the first time used.

Response: Thank you for the suggestion. We have corrected the definition of acronyms in the paper.

9.  P7, L143: Please specify the version of the datasets used, and the link where the data are available.

Response: The following sentence has been added to lines 557-559:

FluxSAT Gross Primary Production is available at https://avdc.gsfc.nasa.gov/pub/tmp/FluxSat_GPP/, while the ALEXI evapotranspiration dataset can be accessed at https://lpdaac.usgs.gov/products/eco3etalexiv001/.

10. P7, L169-170: It is likely that 5-year of spinup is not enough for a correct initialization of the model. In our simulations we typically consider 30-year spinup.

Response: The spinup duration is related to the variable under consideration. For example, if we were analyzing changes in groundwater, then a spinup of five years would have been inadequate. However, in the present case the variable of interest is surface soil moisture. The five-year spin-up is adequate for the present study since soil moisture is a relatively dynamic state variable that rapidly spins up after a small number of precipitation events. The five-year spinup duration was largely motivated by initializing the experiments with a reasonable representation of both the surface soil moisture and the rootzone soil moisture, but was less of a motivating concern with respect to unconfined groundwater.

Additional experiments were carried out for another study in order to test the influence of initial conditions on the subsequent soil moisture conditions and it was noted that a five-year spin-up yielded the same results for the sixth year as did a 10-year spinup across the study domain. In order to comply with standard literature, we have replaced the assimilation results for the five-year spin up with a 10-year spin up. This replacement does not have any influence on the subsequent results though.

11. P9, L185: Where is Table 2? It is the table in Kwon et al. Please check.

Response: Yes, Table 2 is in Kwon et al. We cited that particular table in Kwon et al. to help the readers easily locate the information regarding forcing perturbations within the cited manuscript. The following table has been included in the appendix for the reader's convenience:

**Table B1.** Perturbation parameters applied to meteorological forcing fields for both the open loop and data assimilation simulations. M= multiplicative; A= additive.

| Perturbed meteorological forcing | Perturbation type | Standard deviation | Cross-correlations with perturbations | | | |
|---|---|---|---|---|---|---|
| | | | P | SW | LW | Tair |
| Precipitation (P) | M | 0.5 | - | -0.8 | 0.5 | -0.1 |
| Shortwave radiation (SW) | M | 0.3 | -0.8 | – | -0.5 | 0.3 |
| Longwave radiation (LW) | A | 50 W m$^{-2}$ | 0.5 | -0.5 | - | 0.6 |
| Near surface air temperature (Tair) | A | 1 K | -0.1 | 0.3 | 0.6 | - |

12. P10, L230: It should be clarified the soil layer that is considered for the assimilation of surface soil moisture data, and how the information is propagated with depth.

Response: Surface soil moisture is considered during assimilation since the SMAP retrievals being assimilated represent the top ~5 cm of surface soil. The information is propagated to underlying soil layers based on the water diffusivity and hydraulic conductivity, maximum soil moisture threshold of layers, and moisture flux between subsequent layers of the soil. Noah-MP connects subsequent soil layers such that excessive water above saturation in a layer is moved to the next unsaturated layer similar to a bucket. Further details are provided in (Ek et al., 2003; Niu et al., 2011; Yang et al., 2011).

The following sentence has been added to line 242:
"Since the SMAP retrievals being assimilated represent the top ~5 cm of surface soil, the soil moisture in the topmost soil layer is the model state variable considered during assimilation."

The following sentences have been added at line 108:
"Updates in the surface soil moisture information are propagated to the underlying soil layers based on the water diffusivity and hydraulic conductivity, maximum moisture threshold of soil layers, and moisture flux between subsequent layers of the soil. Noah-MP connects subsequent soil layers such that excessive water above saturation in a layer is moved to the next unsaturated layer."

Ek, M., Mitchell, K., Lin, Y., Rogers, E., Grunmann, P., Koren, V., Gayno, G., and Tarpley, J.: Implementation of Noah land surface model advances in the National Centers for Environmental Prediction operational mesoscale ETA model, Journal of Geophysical Research: Atmospheres, 108, 2003.

Niu, G.-Y., Yang, Z.-L., Mitchell, K. E., Chen, F., Ek, M. B., Barlage, M., Kumar, A., Manning, K., Niyogi, D., Rosero, E., and Tewari, M.: The community Noah land surface model with multiparameterization options (Noah-MP): 1. Model description and evaluation with local-scale measurements, Journal of Geophysical Research: Atmospheres, 116, 2011.

Yang, Z.-L., Niu, G.-Y., Mitchell, K. E., Chen, F., Ek, M. B., Barlage, M., Longuevergne, L., Manning, K., Niyogi, D., Tewari, M., and Xia, Y.: The community Noah land surface model with multiparameterization options (Noah-MP): 2. Evaluation over global river basins, Journal of Geophysical Research: Atmospheres, 116, 2011.

13. P11, L269-270: In Maqu results are different, i.e., RMSE is lower for MERRA2. Please reformulate.

Response: Thank you for the correction. The following sentence has been added to lines 267-270:
"Relative to MERRA2, IMERG-based SM estimates have lower RMSE for the sample location in the Ngari network and higher RMSE for the location in the Maqu network. This

indicates the importance of precipitation boundary conditions in terms of SM estimation across locations of varying climatology (i.e., arid versus humid)."

14. P15, L323: Why overestimation also for Savannas? It should be clarified.
Response: Thank you for your comment. Only 1.4% of the total grid cells included in the study domain belong to the land cover type Savannas. Of the 1.4%, only 40% of the pixels have SMAP retrievals available for assimilation. Therefore, the total instances of assimilation are much lower in number as compared to croplands or baren land cover. Figure 5 describes a general increase in soil moisture magnitudes after assimilation as compared to the OL. Considering the limited number of grid cells used to compute the linear regression coefficient (as compared to shrublands or baren areas), it is difficult to ascertain the exact cause of the generally higher soil moisture magnitudes for the DA-NoCDF estimates relative to the OL. The following sentence has been added to lines 335-337:

"It is difficult to ascertain the exact cause of the generally higher soil moisture magnitudes for the DA-NoCDF estimates relative to the OL for pixels included in savannas due to the small sample size. Approximately 1.4% of the total grid cells included in the study domain belong to the land cover type savannas of which only 40% of the pixels have SMAP retrievals available for assimilation."

15. Figure 6: Due to uncertainties in soil map, I believe this figure is not really useful for the paper and it can be moved to the appendix.
Response: Thank you for your comment. The figure has been relocated to Appendix C under the added section 'OL versus DA estimates with respect to soil texture'.

16. P16, L349: Note that Noah-MP includes an irrigation module that I believe can be very useful for this paper (see also General Comments).
Response: Thank you for the suggestion. Noah-MP does indeed include an irrigation module. A number of our colleagues have studied the utility of the irrigation module included within Noah-MP across South Asia and concluded that the simulation routine was inadequate in representing the general irrigation patterns in this particular domain (Ghatak et al., 2018). In South Asia, irrigation is carried out via a network of formal canals and informal channels and streams. The irrigation network is manually monitored and is therefore quite difficult to explicitly model. Unfortunately, the irrigation module within Noah-MP is unable to accurately replicate these anthropogenic influences on soil moisture via irrigation and is therefore not suitable to adequately model irrigation across the study domain. Zaitchik et al. are currently working on developing a more representative irrigation modeling routine for the study domain that could benefit future explorations of soil moisture across South Asia.

Ghatak, D., Zaitchik, B., Kumar, S., Matin, M.A., Bajracharya, B., Hain, C. and Anderson, M., 2018. Influence of precipitation forcing uncertainty on hydrological simulations with the NASA South Asia land data assimilation system. Hydrology, 5(4), p.57.

17. P24, 479-480: It should be added more details on how irrigation can be quantified using "an inverse method" otherwise I suggest to remove the sentence.

Response: The inverse method is a potential future study. A similar methodology was explored by Brocca et al. (2018) who used coarse-scaled soil moisture retrievals to quantify the amount of water used for irrigation. The following sentences have been added to line 494:

"Brocca et al. (2018) used coarse-scaled soil moisture retrievals to quantify the amount of water used for irrigation. A similar methodology can be explored that uses the difference between the OL and DA estimated soil moisture across croplands to infer information regarding the water quantity supplied by irrigation."

Brocca, L., Tarpanelli, A., Filippucci, P., Dorigo, W., Zaussinger, F., Gruber, A. and Fernández-Prieto, D., 2018. How much water is used for irrigation? A new approach exploiting coarse resolution satellite soil moisture products. International Journal of Applied Earth Observation and Geoinformation, 73, pp.752-766.

---

## Author Response (AR2)

**Author's response**

Dear Dr. Calvet,

Thank you for your comments. We have made our best efforts to modify the manuscript according to the suggestions made by the reviewers. The reviewer comments are included below in black, while the author responses are directly below in red. All the line numbers are according to the revised manuscript. The revised manuscript, supplement document, and track changes document have been uploaded. The track changes document does not support figures. The figures can be viewed in the revised manuscript. Thank you for providing us the opportunity to share our research findings.

Regards,
Ahmad et al.

**Reviewer- 1**

I have read the authors' replies to reviewers comments and the revised paper. Unfortunately, I believe the authors did not address the main problem I have raised in my review, i.e., "the main problem with the paper is that the better results are obtained when SMAP soil moisture data are assimilated without the correction of the BIAS."
The authors replied:
"The main point highlighted in this paper is that CDF-matching removes the irrigation signal from the retrievals, and therefore, we note that better results are obtained across croplands for simulations without any CDF-matching."
My points are:
1) If CDF-matching removes the irrigation signal, the CDF matching approach is wrong and it should be applied differently. I have made a suggestion, but other options can be explored as well.

Response: Thank you for your comments. We have endeavored to update the manuscript according to your suggestions. We found that better results are obtained without any pre-processing of the SMAP soil moisture retrievals (observations). In regions where irrigation is significant, it is recognized that the *a priori* model is biased and that the SMAP retrievals, in general, perform better in these areas. Pre-processing the SMAP soil moisture retrievals (via CDF-matching or some other model-based bias correction procedure) results in the loss of the irrigation signal implicit in these soil moisture retrievals. In turn, such a bias correction step is often detrimental to the soil moisture assimilation exercise.

On the subject of *explicit* representation of irrigation physics within a land surface model, an accurate parameterization of model physics is always preferred over dynamic state updates via data assimilation. If the land surface model was perfect, that would be ideal – and data assimilation would be completely unnecessary. However, an accurate and robust parameterization of anthropogenic irrigation at all locations in space and time does not currently exist. Hence, in the meantime, we present a framework that can help

improve modeled soil moisture estimates when no *explicit* representation of irrigation water quantity or irrigation timing is available. Please see response to comment #2 to reviewer #1 below for more discussion on this particular topic.

To further stress this point, we included results from an alternate CDF scaling method in the supplementary document and provided additional justification as to why a seasonal assimilation methodology would not work for many locations within the study domain. Furthermore, comparison of in-situ measurements across the Tibetan Plateau suggest better performance of assimilation results *without* any pre-processing even though biases are likely present in the land surface model or SMAP retrievals or both. Ideally, in-situ measurements could be used instead of the biased model climatology during the bias correction of the SMAP retrievals. However, the unavailability of publicly accessible soil moisture measurements across croplands (post-2015) limits the use of in-situ measurements for bias correction or validation of the modeled estimates.

We also realize that, although, the DA-NoCDF estimates have better accuracy as discussed in this study, it may not always be the case, i.e., DA-CDF estimates may yield better results for the assimilation of biased satellite retrievals in an unbiased model. Therefore, the implementation of this method may vary depending on the model used, observations assimilated, and the regional hydrologic processes. A series of discussions about the limitations described above, as well as important considerations for a follow-on study, are included at:

Lines 501 to 506 :
"It is important to note that while in the presented study, estimation accuracy is better for assimilation without CDF-matching, the results might be different for other cases. That is, the assimilation of retrievals without bias adjustment may not improve the estimation accuracy as compared to assimilation of CDF-matched satellite retrievals. In this particular study, the SMAP soil moisture retrievals effectively capture the irrigation signal, and as such, help improve the Noah-MP modeled soil moisture estimates via assimilation. However, there is the possibility that assimilation of a different soil moisture retrieval product may degrade the accuracy of the modeled estimates depending on the inherent biases in that given soil moisture retrieval."

Lines 549 to 555:
"Considering the lack of in-situ observations available for use in this study, it is difficult to clearly ascertain the influence of assimilation without CDF-matching in areas that are not irrigated. Across the Tibetan Plateau, DA-NoCDF estimates exhibit the lowest RMSE. However, the evaluation of DA-NoCDF estimates across unirrigated areas in the southern part of the study domain is limited by the scarcity of ground data. A follow-on study should explore the influence of including (as well as excluding) CDF-matching in areas that are not irrigated. This experiment could help explore suitable approaches for incorporating the information obtained from satellite retrievals to correct the modeled estimates without introducing additional bias to the modelled estimates."

Lines 555 to 559:

"In a broader perspective, there is a need to develop a bias correction technique for satellite retrievals that is independent of the accuracy or bias of the model climatology. Using *in-situ* measurements for pre-processing of the satellite retrievals would be one potential method. Current efforts in South Asia by various governmental and non-governmental organizations to measure in-situ soil moisture would benefit the development of suitable methods of bias correction of satellite observations."

2) The same for the model, if the model does not simulate irrigation in an extensively irrigated area, the model is not suitable. For instance, if the model did not simulate evaporation, we surely agree that the model cannot be used. Why not for irrigation?
Response: We agree with the reviewer that it is important that the model used be representative of regional conditions. It is important that the model should include the relevant (dominant) processes that drive the local hydrologic cycle and incorporate explicit treatment of irrigation into the model physics. However, it is quite difficult to accurately model irrigation across South Asia using an explicit framework considering the lack of publicly-reported irrigation rates. In this paper, we show that the soil moisture estimates can be improved through assimilation when using a land surface model that is biased at some locations in space and time. Zhou et al. (2021) are currently working on developing an irrigation modeling routine that would be able to better capture the influence of regional irrigation. This effort is expected to benefit future explorations of soil moisture across South Asia.

The following text has been added to lines 543 to 544:
"Simulating the complex regional irrigation scheme is a difficult task that is further complicated by the inaccessibility of relevant pumping data, manual operation of reservoirs, and unsystematic canal to field irrigation."

Zhou, Y., Zaitchik, B.F., Kumar, S.V. and Nie, W., 2021, December. Satellite-informed simulation of irrigation in South Asia: opportunities and uncertainties. In AGU Fall Meeting 2021, American Geophysical Union.

I apologise the authors for not being more supportive, but I believe that these two points need to be strongly stressed. We should improve our modelling and observation capability, we can't simply assimilate satellite data to correct unmodelled processes. Due to many causes of the bias between model and observations, results can be good or bad by chance.
Response: We agree with the reviewer that it is quite important to improve both the model as well as the observations. All models and observations have certain shortcomings. No dataset perfectly represents regional hydrologic processes, and as such, there will always be a need for corrective updates. However, assimilation provides a way to integrate the benefits of modeling and remote sensing in order to develop a product that has better accuracy than the stand-alone datasets.

The following text has been added to lines 506 to 508:

"It is important that the model physics be improved as well so that the regional hydrologic processes are accounted for, resulting in a more representative model which could then be used for bias correction of satellite retrievals."

**Reviewer- 2**

The authors addressed in detail all the reviewers and editor comments and suggestion. The link between the actual results and discussion/conclusions is now better balanced, reflecting the limitations from the lack of in-situ observations to evaluate the impact of the DA (over the irrigated regions), and the "potentially questionable" use of the satellite data without bias correction. Therefore my recommendation is to accept the manuscript for publication in HESS.
Response: Thank you for your comments. The manuscript has been updated according to the suggested changes.

I just leave a few minor technical suggestion to the authors below.

1.  New title: The new title is more general, but better aligned with the content of the study. Although I'm not native English "via SMAP retrieval assimilation" sounds a bit strange. I would suggest to change it to "Soil moisture estimation in South Asia via assimilation of SMAP retrievals". But I leave this just as a suggestion for the authors. (lines in the document with track changes)
Response: The title has been changes to "Soil moisture estimation in South Asia via assimilation of SMAP retrievals"

2.  Ln 235: "RMSE values were computed for model and SMAP" : "RMSE values were achieved for model and SMAP"
Response: The following text has been added to line 230:
"Based on the test results, it was noted that the smallest bias and RMSE values were achieved for model and SMAP soil moisture retrieval error standard deviations equal to 0.04 $m^3 m^{-3}$."

3.  Ln 523: "would not work in this study": "would have limitations in this region"
Response: The following text has been added to line 513:
"Hence, implementation of CDF-matching only during certain months would have limitations in this region"

4.  Ln 528; Sentence "It highlighted the limitations ... surface model" is not necessary, as this is already mentioned in the first sentence of the paragraph.
Response: The sentence "It highlighted the limitations ... surface model" has been removed from the text.

5.  Ln 533: "new parameters are identified to" to "new parameters are required to"
Response: The sentence at line 521 has been modified to:

"However, an important point to consider is that with an increase in the hydrologic coupling between surface and deep soil layers, the complexity of the land surface model would also increase as new parameters are required to model the feedback loop between adjacent soil layers."

---

## Author Response (AR3)

**Editor's comments**
Dear Authors,

Thanks for the revised version of the manuscript. Could you check the Supplement? In your response you indicate that "To further stress this point, we included results from an alternate CDF scaling method in the supplementary document and provided additional justification as to why a seasonal assimilation methodology would not work for many locations within the study domain" but I was not able to see any difference between the new Supplement and the old version.

Best regards,
JC Calvet.

**Author Response**
Dear Editor,

Thank you for your comments. This sentence refers to the additions made to the supplementary document during the first round of revisions. We added these results to the supplementary document during the first revision and referred to these updates in the response to comments for the second revision (as well). We apologize for any confusion in how we referred to earlier edits in the supplementary document.

Bias correction via anomaly scaling was attempted. It was found that using the model climatology to correct biases using either CDF-matching or anomaly-scaling methods results in the loss of the irrigation signal from the SMAP soil moisture retrievals. In addition, seasonal assimilation would not yield the desired results as the amount of water contributed by irrigation changes in magnitude during different time periods (higher during some months). However, it cannot be quantified as negligible at any time of the year (Biemans et al., 2016). It would be incorrect to assume that irrigation is only important during the winter. Therefore, seasonal CDF-matching would not result in the removal of all the biases from the SMAP retrievals.

Further details regarding the anomaly scaling results are provided at the following lines in the current supplementary document:

Line 12 to 32:
S2: Anomaly-scaled retrieval assimilation
Assimilation using an anomaly-based approach (DA-Anom.) was also tested. In this approach, the retrieval mean was mapped to the land surface model mean and updates were computed using the resultant anomalies such that:

Observed value ($s, t$) = Mean NoahMP soil moisture($s$) + Observation anomaly($s, t$)

where, $s$= location in space and $t$= instance in time. Figure S1 shows a sample timeseries for a location that is 80% irrigation-equipped. It is apparent that assimilation estimates (DA-CDF) after anomaly scaling closely mimic the OL estimated soil moisture throughout

the year whereas DA-NoCDF is able to update the soil moisture based on the information in the SMAP observations, particularly during the winter months.

[Figure]

Figure S1. Comparative timeseries of OL and DA estimated surface (top 5 cm) soil moisture at an irrigation-equipped pixel. The solid line represents the ensemble mean whereas the shaded areas represent mean +/- standard deviation across the full ensemble. DA-CDF: anomaly-based assimilation; DA-NoCDF: no CDF-matching based assimilation.

Figure S2 presents the differences between OL versus DA estimated soil moisture for the two main seasons. DA-CDF (subplots (a) and (e)) and DA-Anom. (subplots (b) and (f)) simulations show some spatial similarities during both seasons. During summer, the DA-Anom. simulations (Fig. S2(b)) do not show any visible updates across the Indus, Ganges, and Brahmaputra basins. This signal is, however, apparent in the DA-NoCDF map (Fig. S2(c)). For winter, the DA-Anom. estimates (Fig. S2(f)) show positive updates across the Ganges Basin, however, little influence is seen across the Indus and Brahmaputra basins.

[Figure]

Figure S2. Differences between the mean soil moisture estimated by the OL and DA simulations during the summer (April 2016 to September 2016) versus the winter months (October 2015 to March 2016). DA-CDF= assimilation of CDF-matched SMAP retrievals; DA-Anom.= assimilation of anomaly scaled SMAP retrievals; DA-NoCDF= no CDF-matching of SMAP retrievals.

Figure S2(d) presents the annual mean differences between the OL and DA-Anomaly runs. Positive differences are observed across the Tibetan Plateau, similar to the DA CDF run (Fig. 4). The statistics show that DA-CDF estimates have the lowest accuracy across the Tibetan Plateau (lower than the OL). The  performance of the individual runs could be further explored if in-situ measurements were available across the lower part of the study domain. Unfortunately, there are no publicly available soil moisture datasets across the three primary river basins in South Asia, i.e., Indus, Ganges, and Brahmaputra, from 2015 onwards.